# GReTo: Remedying dynamic graph topology-task discordance via target homophily

**Zhengyang Zhou**[3], **Qihe Huang**[2], **Gengyu Lin**[2], **Kuo Yang**[3], **Lei Bai**[4], **Yang Wang**[1,2,3*]

[1]Key Laboratory of Precision and Intelligent Chemistry, University of Science and Technology of China (USTC), Hefei, China
[2]School of Software Engineering, USTC. [3]School of Computer Science and Technology, USTC.
[4]Shanghai AI Laboratory, Shanghai, China
{zzy0929,hqh,lingengyu, yangkuo}@mail.ustc.edu.cn,
baisanshi@gmail.com, angyan@ustc.edu.cn*

## Abstract

Dynamic graphs are ubiquitous across disciplines where observations usually change over time. Regressions on dynamic graphs often contribute to diverse critical tasks, such as climate early-warning and traffic controlling. Existing homophily Graph Neural Networks (GNNs) adopt physical connections or feature similarity as adjacent matrix to perform node-level aggregations. However, on dynamic graphs with diverse node-wise relations, exploiting a pre-defined fixed topology for message passing inevitably leads to the aggregations of target-deviated neighbors. We designate such phenomenon as the topology-task discordance, which naturally challenges the homophily assumption. In this work, we revisit node-wise relationships and explore novel homophily measurements on dynamic graphs with both signs and distances, capturing multiple node-level spatial relations and temporal evolutions. We discover that advancing homophily aggregations to signed target-oriented message passing can effectively resolve the discordance and promote aggregation capacity. Therefore, a GReTo is proposed, which performs signed message passing in immediate neighborhood, and exploits both local environments and target awareness to realize high-order message propagation. Empirically, our solution achieves significant improvements against best baselines, notably improving 24.79% on KnowAir and 3.60% on Metr-LA.

## 1 Introduction

Graph-structured data mining has become a popular technique in numerous disciplines, such as social networks (You et al., 2022), road networks (Chen et al., 2020), and molecule analysis (Abu-El-Haija et al., 2019). However, existing solutions to graph mining usually make the assumption of homophily on graphs where connected nodes tend to share similar features or have the same labels (targets). Actually, in real-world graphs, the homophily assumption does not always hold on (Zhu et al., 2020). Thus, Graph Neural Networks (GNNs) considering heterophily are proposed to break the homophily assumption, which disentangle the complex neighborhood components (Ma et al., 2019; Du et al., 2022) and model the edge diversity (Zhu et al., 2021a; Wang et al., 2022a) by separately aggregating similar and dissimilar signals (Bo et al., 2021; Yan et al., 2021). Despite achievements, heterophily GNNs are mostly investigated on classification tasks over static graphs while less explored on node-level regressions over dynamic graphs. Therefore, it provides an opportunity to dissect how edge-type disentanglement boosts regression capacity on dynamic graphs.

Regression tasks are more challenging than classification as the latter only considers discrete labels with much tolerance (Wang et al., 2022b). Actually, nodes in dynamic graphs are more prone to suffer complex neighborhood distributions (Ma et al., 2022) due to the existence of time-varying values and different edge types, incurring misleading message passing when aggregating target-deviated neighbors. The misleading message passing is formally designated as the topology-task discordance in our work (see Fig. 1(a)). We take traffic volumes of road networks as an intuitive example of edge

---

*Prof. Yang Wang is the corresponding author.

diversity in Fig. 1(b)-(c). The neighboring intersections sharing an upstream-downstream connectivity can be positively correlated, while interactions locating parallelly on the same Origin-Destin transition tend to be negatively correlated with a contended relationship. These two-type edges respectively account for homophily and heterophily components and these correlations will also change over time with tidal patterns. Consequently, uniform aggregations on these two-type neighbors will involve interfered noise and deteriorate the performances of GNNs, as not all of them have consistent evolution direction towards targets. Empirically, in four real-world dynamic graphs, both low homophily ratios within intra-graph frames and across temporal adjacent frames (Fig. 1(d)) imply that physically-connected nodes are not necessarily with close observations[1] or with same variation directions. This not only supports the argument of topology-task discordance, also manifests the universality of such phenomenon across different dynamic graphs [2]. Furthermore, due to the heterogeneous local structures and neighborhood distributions, the topology-task discordance can be propagated to high-order neighborhoods. In other words, the optimal receptive fields should be adaptively and efficiently constructed to realize controllable neighborhood aggregation, thus avoiding noise involvement. Therefore, for dynamic graphs, remedying topology-task discordance in both immediate and high-order neighborhoods is urgently desired.

**Challenges.** Based on the spatial-temporal property within node-wise relationships, adapting existing homophily theory to address such topology-task discordance is still challenging. The key obstacles can be summarized as 1) how to determine which pairs of nodes belong to homophily components without categorical labels, 2) how to involve targets to reconstruct node-wise correlations, thus materializing remedied and powerful target-oriented aggregations, 3) how to exploit the dynamic local neighborhood environments to achieve personalized high-order propagations.

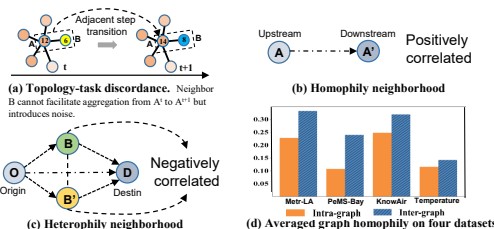

Figure 1: Illustration of edge-type diversity and statistical graph homophily.

**Present work.** Our work empirically and theoretically elucidates the existence of topology-task discordance and explains the failure of homophily GNNs on dynamic graphs. To get rid of such dilemma, we propose a novel GNN to Remedy Topology-task discordance via Target-homophily (GReTo). Firstly, we extend the node-wise relations to a triple tuple based on signed-distance proximity, and construct two measures including intra- and inter-graph homophily to capture diverse spatial-temporal relations and overcome lacking categorical labels. Secondly, by introducing the target awareness with a transition homophily predictor, we incorporate two signed homophily measures to facilitate target-oriented message passing, renewing the activeness of GNNs. Finally, instead of imposing a nested or bi-level inefficient optimization (Xiao et al., 2021), we devise an adaptive layer-wise importance measurement to promote the immediate neighborhood aggregation towards high-order propagations, by identifying the informativeness of each propagation step relative to the expected targets. We evaluate our solution on four dynamic graphs and successfully achieve 3.20% to 24.79% improvements against baselines on MAPE, where KnowAir ($\uparrow$ 24.79%) with higher intra-graph negative heterophily ratios (Tab. 4) especially benefits from flexible signed message passing.

**Contributions.** (1) We formalize a dynamic graph homophily theory, jointly characterizing multi-type node-wise relations considering spatial-temporal property. (2) On dynamic graphs, we analyze the topology-task discordance and corresponding solution from its existence to the solution to personalized high-order propagation. (3) We propose GReTo, consisting of a signed target-oriented message passing and layer-importance based high-order propagation, to refine the topology adapting to downstream regression tasks.

## 2 RELATED WORK

GNNs have become an admirable tool of diverse graph-structured data mining (Song et al., 2022; Kipf & Welling, 2016; Abu-El-Haija et al., 2019). To boost the representation capacity of traditional

---

[1] Observations refer to the observed node features in the graph.

[2] Following graph construction on existing literature (Yu et al., 2018; Guo et al., 2019; Li et al., 2018), we establish the edges between two nodes by selecting top-5% geographically proximal nodes as neighbors.

GCNs, GAT (Veličković et al., 2018) modifies the conventional GNNs by allowing flexible node-level attention, while Directional GNN (Beaini et al., 2021) exploits directional derivatives to endow GNNs with the anisotropic property and overcome the oversmoothing issue. Recently, multiple node-level relations motivate the research trend of homophily theory that measures the consistency between the original topology and node-wise features (or labels) (Zhu et al., 2021b; Bo et al., 2021). These techniques respectively design bi-kernel aggregations (Du et al., 2022; Yan et al., 2021) and high-order neighborhood modeling (Ma et al., 2022; Zhu et al., 2020) to advance the standard GNNs towards more powerful aggregation in addressing heterophily. Besides, researchers carefully devise various high-order propagations by a bi-level learn-to-stop optimization (Xiao et al., 2021) or layer-shared weighted aggregation (Chien et al., 2020). Even flourishing, graph theories mostly focus on classification over static graphs, thus resulting in the opportunity of adapting emerging theories to dynamic graphs. Dynamic graph learning usually adopts RNN backbones (Ruiz et al., 2020; You et al., 2022; Bai et al., 2020) in a spatial-temporal manner, where they construct various adjacent matrices via similarity (Zhou et al., 2020), or learnable embedding product (Wu et al., 2020b). And two pioneering works D2STGNN (Shao et al., 2022) and TAMP-S2GCNET (Chen et al., 2021c), respectively motivated by the dynamic composition separation and time-conditioned topology data, are proposed to capture the spatiotemporal dynamics. Among them, all these works generate the non-negative adjacencies and neglect the influence of evolution trend on topology. However, due to the topology-task discordance induced by node dynamics and edge diversity, these works thus fail to realize the target-oriented aggregations. In contrast, our work investigates the homophily theory in dynamic graphs to capture 'good' neighbors by explicitly modeling the target-related temporal evolution influences. Detailed related works can be found in A.2.

## 3 PRELIMINARIES

Let $\mathcal{T} = \{1, 2, ..., T\}$ be a temporal step set and denote the dynamic graphs as $\mathbb{G} = \{\mathcal{G}_1, \mathcal{G}_2, ..., \mathcal{G}_t, ..., \mathcal{G}_T\}$. Given step $t$, $\mathcal{G}_t = \{\mathcal{V}, \mathbf{X}_t, \mathbf{A}_t\}$ represents the valid observation of graph $\mathcal{G}$. Specifically, the node set $\mathcal{V} = \{v_1, v_2, ..., v_N\}$ is with the cardinality of $N$, and the matrix $\mathbf{A}_t \in \mathbb{R}^{N \times N}$ describes the static weighted adjacency relationship of $\mathcal{G}_t$. Let $F$ be the number of feature dimensions of each node [3], then $\mathbf{X}_t = \{x_1^t, x_2^t, ..., x_N^t\} \in \mathbb{R}^{N \times F}$ serves as the feature matrix. The goal of our dynamic graph regression is to derive an optimized graph learning function $\boldsymbol{F}_g^*$ to predict the observations across the whole graph for the next step, i.e., $\widehat{\mathbf{X}}_{T+1} = \boldsymbol{F}_g^*(\mathbb{G}, \mathbf{X}_t, \mathbf{A}_t | t = 1, 2, ..., T; \boldsymbol{\Theta})$. We name $\{(\mathbf{X}_t, \mathbf{A}_t) | t = 1, 2, ..., T\}$ and $\mathbf{X}_{T+1}$ respectively as the historical and targeted observations, while $\boldsymbol{\Theta}$ refers to all learnable parameters.

### 3.1 DYNAMIC GRAPH HOMOPHILY THEORY

We first introduce the conventional graph homophily ratio in classification (Zhu et al., 2020; Ma et al., 2021), which is defined as the fractions of nodes in their neighborhoods that with the same label $y_i$ as $v_i$, i,e., $ho_i = \frac{\sum_{v_j \in \mathcal{N}(v_i)} \mathbb{I}(y_i = y_j)}{|\mathcal{N}(v_i)|}$. Here, $\mathcal{N}(v_i)$ refers to the neighboring node set of $v_i$. In contrast, we propose our dynamic graph homophily theory to adapt the continuous data space. Our dynamic graph homophily theory includes descriptors of homophily measurements and local neighborhood environments, which are two factors of node-level and neighborhood-level statistics that inherently affect the information propagation. Concretely, we first exploit signed distances to extend the binary relations to a triple tuple, compensating for non-explicit class boundaries. For dynamic graphs, we formulate an intra-graph spatial homophily to capture node correlations in same graph frames and an inter-graph transition homophily to extract temporal evolution between adjacent temporal frames. Second, to numerically depict the neighborhood on both structures and neighbor compositions, we integrate the topological feature (node degree $d_i$) with homophily distribution as local neighborhood environments.

***Definition 1 (Intra-graph spatial homophily and Inter-graph transition homophily)*** *Both intra- and inter-graph homophily can be categorized into three classes according to the feature distances by a proximity threshold $\varepsilon$, with the signs indicating the direction of proximity. Given a serial graphs $\mathbb{G}$, temporal step $t$ and node $v_i$, the intra-graph homophily $(\pi_{ij}^s)_t$ and inter-graph homophily $(\pi_{ij}^T)_t$*

---

[3] We set $F$=1 but it is orthogonal to our theory as it can be easily extended to tensor formats.

*are respectively defined by a symmetric formation (see Fig. 5(a)) as,*

$$(\pi_{ij}^s)_t = \begin{cases} -1, \frac{||x_j^t||-||x_i^t||}{||x_i^t||} < -\varepsilon \\ \varepsilon, \frac{||x_j^t-x_i^t||}{||x_i^t||} \le \varepsilon \\ 1, \frac{||x_j^t||-||x_i^t||}{||x_i^t||} > \varepsilon \end{cases} \quad (\pi_{ij}^T)_t = \begin{cases} -1, \frac{||x_j^{t+1}||-||x_i^t||}{||x_i^t||} < -\varepsilon \\ \varepsilon, \frac{||x_j^{t+1}-x_i^t||}{||x_i^t||} \le \varepsilon \quad (v_j \in \mathcal{N}(v_i)) \\ 1, \frac{||x_j^{t+1}||-||x_i^t||}{||x_i^t||} > \varepsilon \end{cases} \quad (1)$$

*In particular, the signed direction indicates how will the node value change when it aggregates a specific neighbor in $\pi_{ij}^s$, while the direction demonstrates how the node is expected to change to reach the target in $\pi_{ij}^T$.*

**Definition 2 (Local neighborhood environments (LNE))** *LNE consists of local structure descriptions and neighborhood distributions. Given node $v_i$, we let the node degree $d_i$ describe the local structure and further partition the nodes of its neighborhood $\mathcal{N}(v_i)$ into three groups based on node-wise feature distances. In detail, the neighboring components those are proximal, positively deviated, and negatively deviated away the central node $v_i$ are respectively considered as spatial homophily $(\mathcal{N}(v_i))_s$, positive heterophily $(\mathcal{N}(v_i))_{qp}$, and negatively heterophily $(\mathcal{N}(v_i))_{qn}$. We can transfer the neighborhood components into a numerical* Neighborhood Distributions *vector $\boldsymbol{\pi}_{v_i}$,*

$$\boldsymbol{\pi}_{v_i}(p_s, q_p, q_n) = \left[ \frac{\sum_j \mathbb{I}(\pi_{ij}^s = \varepsilon)}{|\mathcal{N}(v_i)|}, \frac{\sum_j \mathbb{I}(\pi_{ij}^s = 1)}{|\mathcal{N}(v_i)|}, \frac{\sum_j \mathbb{I}(\pi_{ij}^s = -1)}{|\mathcal{N}(v_i)|} \right] (v_j \in \mathcal{N}(v_i)) \quad (2)$$

*where $(p_s, q_p, q_n) = \{p(\pi_{ij}^s = \varepsilon), p(\pi_{ij}^s = 1), p(\pi_{ij}^s = -1)\}$. Then the local neighborhood environment can be denoted as $\boldsymbol{LNE}_{v_i} = (d_i, p_i, q_{p_i}, q_{n_i})$.*

## 4 DISCORDANCE BETWEEN TOPOLOGY AND TASKS

In this section, we show the discordance between the topology and task, when adopting traditional homophily GNNs. Given a serial graphs $\mathbb{G}$ and node $v_i$, (1) the ratio of intra-graph spatial homophily $\boldsymbol{\pi}_{v_i}(p_s)$ is simplified to $p_i$, while the ratio of other node-level relations is denoted as $q_i = 1 - p_i$, (2) the target coefficient $\gamma_i$ quantifies the relationship between the targeted step $T + 1$ and a given step $t$ by $x_i^{T+1} = \gamma_i x_i^t$. Our analysis only considers aggregations from $T$ to $T + 1$, and let $x_i^{T+1} = \gamma_i x_i^T$. We ignore the temporal superscript for simplification in following analysis. Actually, given a target coefficient $\gamma_i$, a successful aggregation can be equivalent to finding a valid $d_i$ to construct an appropriate neighborhood for achieving targets. Therefore, we establish the relationship between $\gamma_i$ and degree $d_i$, to find what conditions will the homophily-assumed graph aggregation fail.

**Theorem 1 (Proof in Appendix B.1.)** *Consider impose a homophily GNN $\boldsymbol{F}_g$ on $\boldsymbol{X}$ for node-level regressions, the relationship between the expected degree $d_i$ and the target coefficient $\gamma_i$ is,*

$$d_i = \begin{cases} (-\infty, -1), \ \gamma_i < p_i \\ \infty, \gamma_i = p_i \\ [0, +\infty), \ p_i < \gamma_i \le 1 \\ (-1, 0), \ \gamma_i > 1 \end{cases} \quad (3)$$

Theorem 1 demonstrates that homophily aggregations cannot find the optimal neighborhoods to achieve the targeted results when $\gamma_i < p_i$ and $\gamma_i > 1$. In other words, the homophily GNNs will fail when the gap between historical observations and targeted prediction value is large.

## 5 REMEDY TOPOLOGY-TASK DISCORDANCE ON DYNAMIC GRAPHS

In this section, we propose our solution, GReTo, to remedy the topology-task discordance in dynamic graph regressions, where it consists of Signed target-oriented message passing, and Personalized high-order layer propagation. The framework overview is illustrated in Fig. 2.

### 5.1 SIGNED TARGET-ORIENTED MESSAGE PASSING

**Overview.** We first provide a theoretical analysis on the potential of aggregation capacity improvement by exploiting homophily theory. Then we propose to design a signed target-oriented message passing, which captures the task-oriented evolution direction to remedy the dynamic graph topology.

**Theorem 2 (Proof in Appendix B.2.)** *Given a central node $v_i$, and target coefficient $\gamma_i < p_i$ or $\gamma_i > 1$, the aggregation capability can be improved by a sign-preserved directional message passing that separately aggregates positive and negative heterophily compositions. Under such message passing, we denote $s_p > 0$ and $s_q < 0$ as the signed kernels, then the aggregation algorithm can conditionally converge to $x_i^{T+1}$ where $d_i$ satisfies the following two conditions,*

$$
\begin{cases}
E[h_i^T] - x_i^{T+1} = \dfrac{x_i + d_i p_i (1 \pm \widetilde{\varepsilon}) x_i + s_p \sum\limits_{j \in N_+(v_i)} w_p x_j + s_q \sum\limits_{k \in N_-(v_i)} w_q x_k}{d_i + 1} - \dfrac{\gamma_i x_i (d_i + 1)}{d_i + 1} = 0 \\
d_i \in \mathbb{Z}^+
\end{cases}
$$

(4)

*where the expectation of neighbors with intra-graph homophily denotes $d_i p_i (1 \pm \varepsilon) x_i$. $N_+(v_i)$ and $N_-(v_i)$ denote the positively and negatively heterophily neighborhoods, and $w_p, w_q$ are learnable weights. Detailed conditions for obtaining targets are demonstrated in Appendix B.2. With these derived conditions, we can estimate whether the GNN can successfully aggregate the neighbors towards targets when the dataset is given.*

Therefore, the key to our target-oriented message passing becomes investigating both intra- and inter-graph homophily to discover the target-beneficial neighborhoods, and devising meaningful directional aggregation. The technical stages are three-fold, 1) discover the target-oriented homophily neighborhoods, 2) predict node-wise inter-graph transition homophily to enable target awareness, and 3) disentangle different types of neighborhoods for target-aware aggregations.

Figure 2: Framework overview of GReTo.

**Target-homophily neighborhood discovery.** To explicitly distinguish target-oriented neighboring nodes, we resort to our dynamic homophily theory with both signed direction and proximity. Considering that the signs in these two homophily measurements indicate the directional consistency, we couple the spatial one and transition one to derive consistency between the intra-graph connections and the targeted graph. The signed target-oriented neighborhood disentanglement matrix $\widehat{M}$ can be achieved by an element-wise division $\oslash$, i.e., $\widehat{M}_{ij} = \widehat{\pi_{ij}^T} \oslash \pi_{ij}^s$, where $\widehat{\pi_{ij}^T}$ is the estimated transition homophily that will be predicted in the next subsection. This design of division enables each $\widehat{M}_{ij} \in \{1, -1, \varepsilon, -\varepsilon, \frac{1}{\varepsilon}, -\frac{1}{\varepsilon}\}$ in $\widehat{M}$ to possess informative edge descriptions, i.e., each edge value can not only discriminate the target-oriented nodes (with positive signs) as good neighbors, but also measure the degree of node-wise proximity where larger values imply more target-oriented consistency and contribution. By splitting $\widehat{M}$ based on the edge signs, we can achieve the target-oriented homophily and heterophily neighbors by,

$$\widehat{M} = [\widehat{M}]_P + [\widehat{M}]_N \tag{5}$$

In Eq 5, $[\widehat{M}]_P$ is the disentangled aggregation matrix with positive elements interpreting the target-homophily adjacent connections, while $[\widehat{M}]_N$ composes negative values explaining the target-heterophily relationship. Noted that although the original triple relations are shrunk into the signed matrix for efficient implementations, the directional proximity indicating the informativeness for aggregation towards targets is carefully preserved by diverse values in $\widehat{M}$.

**Inter-graph transition homophily predictor.** Since the transition homophily is not available during inference stages, we formulate a sequence classification task to predict which type of relation that the transition homophily belongs to. However, computations among node-level transition homophily are inefficient, we thus reduce the transition homophily to a self-transition homophily between adjacent steps $Sig_i^t = (\pi_{ii}^T)|_{t-1 \to t}$, which can be available from historical observations. Then, we leverage the nice property of periodicity and continuity in time-series to construct the sequence learning. This time-series learner concatenates two parallel LSTMs and one Conv1D to capture the evolution and trend dependence by taking step-wise historical node observations and self-transition homophily $[X_t; Sig^t](t = 1, 2, .., T)$ as inputs. Thus, we obtain the estimated next-step representation by,

$$\widehat{Sig}_R^{T+1} = \text{Conv1D}(\text{LSTM}(X_1, X_2, ..., X_T); \text{LSTM}(Sig^1, Sig^2, ..., Sig^T)) \tag{6}$$

We then utilize the time-series representation $\widehat{\boldsymbol{Sig}}_R^{T+1}$ for a three-category classification where the class probability is estimated with an MLP parameterized by $\boldsymbol{W}_s$, i.e.,

$$[\widehat{\boldsymbol{p}},\ \widehat{\boldsymbol{q}}_p,\ \widehat{\boldsymbol{q}}_n]^{T+1} = \mathrm{MLP}(\widehat{\boldsymbol{Sig}}_R^{T+1}; \boldsymbol{W}_s) \tag{7}$$

The estimated $\widehat{\boldsymbol{p}},\ \widehat{\boldsymbol{q}}_p,\ \widehat{\boldsymbol{q}}_n$ refer to probabilities of three homophily categories that $\widehat{\boldsymbol{Sig}}_R^{T+1}$ belongs to, and then we can obtain the final $\widehat{\boldsymbol{Sig}}^{T+1}$ based on these probabilities.

**Disentangled signed directional message passing.** With modified transition homophily $\boldsymbol{Sig}^{T+1}$, the disentangled target-oriented matrix can be altered as $\widehat{M}_{ij} = \widehat{Sig}_i^t \oslash (\pi_{ij}^s)^t$. To accommodate separate aggregations over different types of node-wise connections, we take advantage of the positive target-homophily matrix $[\widehat{\boldsymbol{M}}]_P$ and negative target-heterophily matrix $[\widehat{\boldsymbol{M}}]_N$ to respectively filter the adjacence $\widehat{A}_i$ where we impose a self-loop on the adjacence $\widehat{\boldsymbol{A}} = \boldsymbol{A} + \boldsymbol{I}$. Thus, the two aggregation kernels can be realized as,

$$(\boldsymbol{f}_L)_i = ([\widehat{M}]_P)_i \odot \widetilde{A}_i, \ \ (\boldsymbol{f}_H)_i = ([\widehat{M}]_N)_i \odot \widetilde{A}_i \tag{8}$$

where $\odot$ is an element-wise Hadamard Product for kernel filtering. Considering the relational property of node-wise signals over graphs, $\boldsymbol{f}_L$ and $\boldsymbol{f}_H$ are respectively interpreted as a low-pass filter and a high-pass filter in graph convolutions (Bo et al., 2021). Then the disentangled signed directional message passing can be formulated as,

$$\mathrm{AGGR}(v_i) = \alpha \sum_{j \in \mathcal{N}_L(v_i)} (\boldsymbol{f}_L)_{ij} x_j w_p + (1 - \alpha) \sum_{k \in \mathcal{N}_H(v_i)} (\boldsymbol{f}_H)_{ik} x_k w_n \tag{9}$$

where $\mathcal{N}_L(v_i)$ and $\mathcal{N}_H(v_i)$ denote the target-homophily and target-heterophily neighbor set regarding $v_i$, and $\alpha$ adjusts the proportion between two compositions. In addition, $w_p \in \boldsymbol{W}_P$ and $w_n \in \boldsymbol{W}_N$ are two learnable parameters for feature transformation. In Eq. 9, the negative kernels can make sense when the aggregated neighbors are exactly negatively correlated with targets, which potentially closes the gap between historical node observations and targeted ones. Thus, our strategy can explicitly model multi-type edges and boost our aggregation power.

## 5.2 Personalized high-order layer propagation

**Overview.** Since the propagation step is a discrete value, it is intractable to directly optimize such discrete values. In this subsection, we exploit the local neighborhood environments to quantify the informativeness of node representations at each propagation step, which determines when to stop the propagation in a soft manner. Concretely, our high-order propagation is with two components, an adaptive layer importance measure and high-order propagation blocks.

**Relationship between local neighborhood environments and propagation steps.** Given node $v_i$, (1) the optimized propagation step is determined by the proportion of target-homophily neighborhoods $\widetilde{p}_i$ (rather than intra-graph homophily $p_i$) and the substructure description $d_i$. (2) the smaller values of $d_i$ and $\widetilde{p}_i$ are, the larger propagation step $K$ is expected. By denoting $INFO(h_i^k)$ as the target-oriented information encapsulated in the representation of the $k$-th propagation step in message passing, Theorem 3 demonstrates the quantitative relationship between the informativeness of each propagation layer and the local neighborhood environments, to verify our qualitative analysis.

***Theorem 3 (Proof in Appendix B.3)*** *Consider a series of dynamic graphs $\mathbb{G}$ that conform to the neighborhood distribution $\mathcal{N}(v_i) \sim \boldsymbol{\pi}_{v_i}(p, q_p, q_n)$ where we enforce $q_n = \eta q_p$, $p + q_p + \eta q_p = 1$. Consider $x_i^{T+1} = \gamma_i x_i^T > x_i^T$ with $\gamma_i > 1$, the quantitative relationship among the layer-specific target-oriented information $INFO(h_i^k)$, substructure $d_i$, and the target-homophily $\widetilde{p}_i$ is as follows. Under this case, the expected target-homophily compositions of neighborhoods $\widetilde{p}_i$ becomes $q_n$, and the difference between the probability of hitting beneficial nodes in the second layer and the first layer $\Delta P(N_+)|_{1-2}$ is $(-1 - 2\eta)q_p^2 + q_p$. We have,*

$$\begin{cases} INFO(h_i^2) > INFO(h_i^1), & 0 < q_p < \frac{1}{1+2\eta} \\ INFO(h_i^1) \geq INFO(h_i^2), & q_p \geq \frac{1}{1+2\eta} \end{cases} \tag{10}$$

*The symmetric results can be achieved when we have a small $\gamma_i$ satisfying $\gamma_i < 1$.*

Theorem 3 verifies that in dynamic graphs with complex neighborhood distributions, lower node-level activeness and target-oriented homophily must result in larger receptive fields and vice versa. Therefore, we are expected to pick the high-order neighbors that are target-homophily.

**Adaptive layer importance measurement.** To pick the high-order target-homophily neighbors, we transfer this challenge to measuring the consistency between high-order neighborhood distributions and transition homophily distribution of targets. First, we exploit the edge composition $\boldsymbol{\pi}_{v_i}$ to construct a neighborhood distribution matrix $\boldsymbol{ND}_h = NeighH(\boldsymbol{X}) = [\pi_{v_1}; \pi_{v_2}; ...; \pi_{v_N}] \in \mathbb{R}^{N \times 3}$ where $NeighH(\boldsymbol{X})$ is a neighborhood disentanglement function on graph-wide features $\boldsymbol{X}$. Then $\boldsymbol{ND}$ in the $k$-th propagation step is $\boldsymbol{ND}_h^k = NeighH(\boldsymbol{A}^k \boldsymbol{X})$. Second, since dynamic graphs experience continuous temporal evolution, the variations of consecutive steps can imply fine-grained fluctuations that progressively approach targets. Therefore, for each $t$, we exploit the estimated transition probability distribution on three categories predicted by Eq. 15 to construct the step-wise transition distribution matrix, i.e., $(\boldsymbol{ND}_T)_i^t = [\widehat{\boldsymbol{p}}, \widehat{\boldsymbol{q}}_p, \widehat{\boldsymbol{q}}_n]^t \in \mathbb{R}^{N \times 3}$. As a result, $(\boldsymbol{ND}_T)_i^t$ naturally encapsulates the target information. Then, we can measure the consistency between the expected $k$-th order neighborhood and the target-oriented transition homophily probability by imposing a Hadamard Product to $(\boldsymbol{ND}_h^k)_i^t$ and $(\boldsymbol{ND}_T)_i^t$. Then a *MAX* operation is applied in each row to preserve the maximal result and suppress the non-target-oriented noise. Besides, a smaller degree tends to encourage aggregating nodes of farther horizons while larger propagation layer should be less weighted. Thus, the layer importance measurement $\phi_i^k$ for node $v_i$ at layer $k$ can be computed by,

$$\phi_i^k = LiM((\boldsymbol{ND}_h^k)_i, (\boldsymbol{ND}_T)_i) = \frac{d_i^k \cdot MAX((\boldsymbol{ND}_h^k)_i \odot (\boldsymbol{ND}_T)_i)}{k} \tag{11}$$

where $d_i^k$ denotes the averaged degree of nodes in $v_i$'s $k$-th order neighborhood.

$$\phi_i^k = \frac{d_i^k \cdot MAX((\boldsymbol{ND}_h^k)_i \odot (\boldsymbol{ND}_T)_i)}{k} \tag{12}$$

**Personalized high-order propagation block.** The core technique of high-order layer propagation is learning a set of interpretable node-specific soft weights for aggregating multi-order neighbors. To achieve such well-normalized aggregation weights, we impose Softmax on the estimated layer importance $\phi_i^k$, associated with an MLP parameterized by $\boldsymbol{W}_{lim}$ to enable it learnable, i.e., $\widetilde{\phi_i^k} = \text{MLP}(LiM((\boldsymbol{ND}_h^k)_i, (\boldsymbol{ND}_T)_i), \boldsymbol{W}_{lim})$. Given maximal propagation step $K$, we can obtain the soft weights over layer-wise aggregations $\boldsymbol{\psi} \in \mathbb{R}^{N \times K}$ by $\boldsymbol{\psi}_i = \text{Softmax}\left[\widetilde{\phi_i^0}, \widetilde{\phi_i^1}, ..., \widetilde{\phi_i^{K-1}}\right]$ where $\boldsymbol{\psi}_i$ is the $i$-th row in $\boldsymbol{\psi}$. So far, we can build our high-order propagation blocks. As target-heterophily compositions are less important during aggregations, we only consider the high-order propagation over target-homophily neighborhoods while imposing one-layer aggregation on target-heterophily components. Then the $K$-th layer representation $\boldsymbol{H}^{(K)}$ can be achieved by,

$$\boldsymbol{H}^{(K)} = \vec{\boldsymbol{\alpha}} \sum_{k=0}^{K-1} \boldsymbol{\psi}_k \boldsymbol{f}_l^k \boldsymbol{X} \boldsymbol{W}_p + (\mathbf{1} - \vec{\boldsymbol{\alpha}}) \boldsymbol{f}_h \boldsymbol{X} \boldsymbol{W}_n \tag{13}$$

There are two distinctions of our designs. First, instead of tuning parameters, we instantiate an $\vec{\boldsymbol{\alpha}} \in \mathbb{R}^N$ as a learnable gate to reduce the fine-tuning burden, i.e., $\vec{\boldsymbol{\alpha}} = \sigma(\text{MLP}(\boldsymbol{H}^{(K)}; \boldsymbol{W}_{mlp}))$ and adjust the compositions between target-homophily and target-heterophily. $\boldsymbol{W}_{mlp} \in \mathbb{R}^{2N \times N}$ are learnable parameters. Second, rather than a complicated routing mechanism, we exploit a learnable but interpretable matrix to reweight the importance of each layer specific to nodes, contributing to efficient high-order message passing. We plug our layer propagation into every temporal step.

## 5.3 Temporal inception layer and optimization

**Temporal inception layer.** To efficiently capture the temporal evolution, we adopt a fully-convolution based sandwich structure as the temporal inception layer, which is inherited from STGCN (Yu et al., 2018). We can formulate the temporal inception layer as $\boldsymbol{H}^{(K)} = \boldsymbol{\Gamma} *_\kappa (\boldsymbol{F}_g^*(\widetilde{\boldsymbol{X}}))$ and denote the final outputs as $\widehat{y}_i \in \widehat{\boldsymbol{Y}}$. Details can be found in Appendix A.1.4.

**Training objective.** The optimization objective of GReTo composes of two parts, the transition homophily estimation implemented by a cross-entropy-based multi-class classification, and the dy-

Table 1: Performance comparisons on various baselines.

| | Metr-LA | | PeMS-Bay | | KnowAir | | Temperature | |
|---|---|---|---|---|---|---|---|---|
| | MAPE | RMSE | MAPE | RMSE | MAPE | RMSE | MAPE | RMSE |
| GCN | 0.0975 | 8.3098 | 0.0522 | 4.4952 | 0.3146 | 16.5635 | 0.3221 | 1.4439 |
| GAT | 0.0628 | 5.8018 | 0.0176 | 1.6610 | 0.2435 | 13.3114 | 0.3393 | 1.4855 |
| GraphSAGE | 0.0606 | 5.7550 | 0.0167 | 1.6173 | 0.2449 | 13.1932 | 0.1966 | 1.0233 |
| SuperGAT | 0.0623 | 5.7886 | 0.0175 | 1.6606 | 0.2535 | 13.3671 | 0.3224 | 1.3439 |
| EGConv | 0.0609 | 5.7554 | 0.0167 | 1.6139 | 0.2399 | 13.2189 | 0.1875 | 1.0097 |
| $H_2$GCN | 0.0608 | 5.7292 | 0.0168 | 1.6599 | 0.2371 | 13.1207 | 0.1906 | 0.9971 |
| STGCN | 0.0554 | 3.8655 | 0.0197 | 1.5890 | 0.2437 | 12.3601 | 0.1704 | 1.1190 |
| GWN | 0.0528 | 3.8434 | 0.0163 | 1.5482 | 0.2288 | 12.8495 | 0.1607 | 0.9132 |
| MTGNN | 0.0526 | 3.8153 | 0.0170 | 1.5759 | 0.2271 | 12.9091 | 0.1682 | 0.9034 |
| DCRNN | 0.0532 | 3.8798 | 0.0161 | 1.5292 | 0.2392 | 13.0389 | 0.1351 | 0.9715 |
| ASTGNN | 0.0530 | 5.5313 | 0.0169 | 1.6229 | 0.2485 | 13.2274 | 0.2978 | 0.9330 |
| GReTo (Ours) | 0.0500 | 3.6552 | 0.0166 | 1.4813 | 0.1708 | 11.0369 | 0.1341 | 0.8704 |

namic graph regression based on MSE. The final objective can be written as,

$$Loss(\boldsymbol{\Theta}) = -\frac{1}{N}\sum_{i=1}^{N}\{\sum_{\substack{p*\in\{p,\\q_p,q_n\}}}(p_i^*\log\widehat{p_i^*} - (1-p_i^*)\log(1-\widehat{p_i^*}))\} + \frac{1}{N}\sum_{i=1}^{N}(\widehat{y_i} - y_i)^2 \qquad (14)$$

## 6 EXPERIMENT

### 6.1 IMPLEMENTATION DETAILS

**Dataset description.** We collect four cross-domain dynamic graph datasets. All of them possess the mixed property of homophily and heterophily. **Traffic: (1) Metr-LA**: Highway traffic status consisting of 207 loop detectors of Los Angeles (Li et al., 2018). **(2) PeMS-Bay**: Traffic statuses collected by California Transportation Agency, including 325 sensors in Bay Area (Li et al., 2018). **Climate: (3) KnowAir** : $PM_{2.5}$ Concentrations, covering 184 main cities in China (Wang et al., 2020). **(4) Temperature**: Urban Temperatures of the same 184 cities as KnowAir (Wang et al., 2020). We construct the graphs based on geographical distances where the node values and virtual topology can be varied across temporal steps. The $\delta\%$ is set as 5% across all datasets to control the proportion of node connections. Dataset details are elaborated in Table 3 and 4.

**Protocols.** We evaluate GReTo on the dynamic graph regression tasks and compare it with state-of-the-art graph architectures. Training. We alternately train our model, i.e., train the transition homophily predictor by freezing other weights while utilizing the well-learned transition homophily predictor to continue training the remaining architecture, where Adam SGD (Kingma, 2014) is the optimization strategy. We initialize the learning rate of 1e-3 with a weight decay 0.99. Evaluations. We employ classic GNNs, topology-refined GNNs, heterophily GNN and spatiotemporal networks for comparisons, and adopt MAPE/RMSE to jointly evaluate the regression performances. Parameters. The homophily criteria $\varepsilon$ are set as 0.08, 0.10, 0.10 and 0.12, while the maximal propagation steps $K$ are set as 6, 6, 3, 4 on MetrLA, PeMS-Bay, KnowAir and Temperature, according to empirical evaluations. The size of TCN kernels is set to $1*3$ on all datasets. For fairness, for each compared GNN, the number of hidden layers is set to 6 and the hidden dimension for each GCN is set to 64. For GAT, the number of heads is set to 8 according to the default setting in their papers.

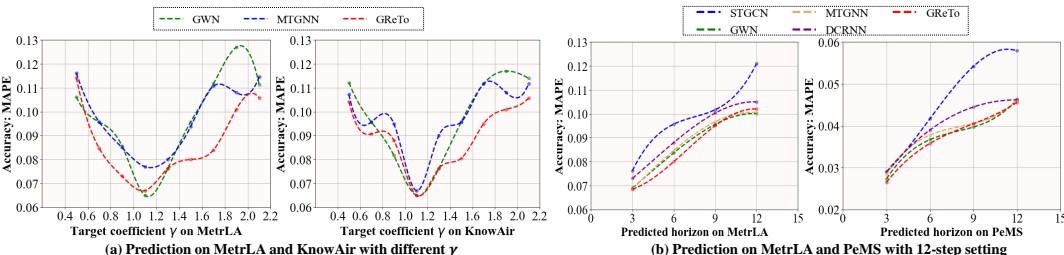

Figure 3: Detailed evaluations on different $\gamma$ and multi-step prediction.

**Baseline.** We compare our solutions with respective four types of baselines. **[1] Classic GNN models:** (1) GCN (Kipf & Welling, 2016), (2) GAT (Veličković et al., 2018), (3) GraphSAGE (Hamilton et al., 2017). **[2] Topology refined models:** (4) SuperGAT (Kim & Oh, 2020). (5) EGConv (Tailor

et al., 2021). **[3] Heterophily GNN:** $H_2$GCN (Zhu et al., 2020). **[4] Spatiotemporal graph learning:** (1) STGCN (Yu et al., 2018). (2) MTGNN (Wu et al., 2020b). (3) GraphWaveNet (GWN) (Wu et al., 2019). (4) DCRNN (Li et al., 2018). (5) ASTGNN (Guo et al., 2019).

**Accuracy of transition homophily prediction.** We present the Classification Accuracy between the estimated ones and groundtruth, i.e., 0.78/0.60/0.86/0.68 on respectively four datasets (1)-(4). As observed, the higher intra-graph homophily tends to lead the higher prediction accuracy and we can also obtain a barely satisfactory result on PeMS and Temperature, which probably benefits from the data property of proximity and continuity. These promising accuracy can imply the feasibility of our transition homophily predictor and ensure the rationality of following implementations.

**Performance comparison.** Comparison results are figured in Table 1. The best result is in bold and the second best is underlined. Our solution almost beats all SOTA baselines, achieving promising performances on four datasets with both 2 metrics. Promisingly, GReTo improves MAPE by 24.79% and 3.61% respectively on KnowAir and MetrLA, where KnowAir with high negative heterophily $q_n$ benefits from flexible signed message passing instead of only similarity-based aggregation. In detail, classic GNNs fail to involve the transition-oriented temporal information approaching real targets, reasonably resulting in large performance margins between theirs and our GReTo, also inferior to other spatiotemporal networks. Generally, EGConv and H2GCN obtain better results and we can observe that the higher heterophily of the graphs is, the more superior performances of GReTo, EGConv and H2GCN reveal. The underlying distinctions lie in the spatially-varying adaptive filters of EGConv and heterophily tolerance of H2GCN, supporting our idea of target-homophily topology refinement in dynamic graphs. Extensive comparisons illustrate that our work advances the interpretation and designs of graph-based theory forward spatiotemporal learning and benefits learning on graphs that are with dynamic topology and heterophily compositions.

**Ablation study.** (1) Ablate the target-oriented neighborhood calibration, reducing to traditional GCN (V-TO). (2) Replace the high-order layer propagation with layer-wise weighted fusion, reducing to Mixhop (V-Mix). (3) Reduce the high-order layer propagation to a uniform multi-layer fusion with the same weight (V-ULW). We report

Table 2: Performance comparisons (MAPE) on ablative variants

| Datasets | Metr-LA | PeMS-Bay | KnowAir | Temperature |
|---|---|---|---|---|
| V-TO | 0.0805 | 0.0430 | 0.2891 | 0.2357 |
| V-Mix | 0.0516 | 0.0176 | 0.1799 | 0.2523 |
| V-ULW | 0.0519 | 0.0180 | 0.1800 | 0.2434 |
| **GReTo** | **0.0500** | **0.0166** | **0.1708** | **0.1341** |

MAPE on four datasets in Table 2. Overall, promising results confirm the effectiveness of the two well-designed modules. In particular, target-oriented designs play significant roles across all datasets with the improvement ranging from 37% to 61%, while the personalized high-order propagation reveals prominent improvements on Temperature. Regarding the fusion strategy, the layer-wise weighted fusion and uniform aggregation share similar performances while our node-specific personalized propagation shows a remarkable promotion. Noted that these ablation results also show that two modules are with diverse sensitivity to datasets with different properties such as graph sizes or graph homophily, but our GReTo is a general solution revealing promotions across four datasets.

**Generalization on different $\gamma$ and multi-step prediction.** We respectively select two datasets for a detailed study. First, we derive the statistical MAPE at different $\gamma$ ranging from 0.5 to 2.1. Results are shown in Figure 3(a). The performances reveal a 'V' shape with $\gamma$ increasing, indicating better results when $\gamma$ approaches 1 while inferior performances as it deviates away from 1. Similarly but promisingly, our solution exhibits lower errors and more robustness, which can be attributed to signed target-oriented message passing. Second, we present our multi-step performances against several representative baselines by modifying our framework to a 12-step forecasting with a sequential output where results are in Figure 3(b). We observe that longer horizons enforce the target-oriented design to gradually lose ground as our GReTo does not tailor for multi-horizon settings. Even so, our solution still achieves comparable performances with best baselines, which verifies the generalization and potential of GReTo in modeling longer horizons.

## 7 CONCLUSION

In this paper, we propose a dynamic homophily theory by revisiting node-wise relationships from spatial-temporal perspectives and show the opportunity to renew aggregation capacity of conventional GNNs. Technically, a novel GNN, i.e., GReTo is proposed, which integrates the signed target-oriented message passing and the layer importance based propagation, to refine the topology in both immediate and high-order neighborhoods. Empirical experiments have validated the superiority of

our GReTo on four dynamic graphs. We will continue to work on how homophily theory improves generalized dynamic graph learning, e.g., edge-type prediction and multi-step series predictions. Our code is available at https://github.com/zzyy0929/ICLR23-GReTo.

## 8 ACKNOWLEDGEMENT

This work is partially supported by the National Natural Science Foundation of China (No.62072427, No.12227901), the Project of Stable Support for Youth Team in Basic Research Field, CAS (No.YSBR-005), Academic Leaders Cultivation Program, USTC.

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

# A  APPENDIX

## A.1  MORE DETAILS ABOUT PROPOSED GRETO

### A.1.1  MOTIVATION OF GRETO

In general, we discover that some topologically connected neighbors sometimes tend to be dissimilar with targets and the aggregation of these target-deviated neighbors will introduce noise and harm the performance. In dynamic graphs, the node values will change over time thus the node-wise relationship, i.e., homophily on graphs, will also accordingly tend to be time-varying. To this end, the intuition of this work is that not all neighbors are beneficial to aggregate for achieving targets, especially in dynamic graph regression tasks, where we designate such phenomenon as topology-task discordance. Actually, only target-oriented neighbors are beneficial and encouraged to be aggregated. Therefore, inspired by the homophily theory in static graphs, we extend the graph homphily theory to dynamic graph scenarios and highlight the beneficial nodes based on fine-grained analysis of neighborhood compositions to remedy the topology-task discordance. We further design layer-wise importance measures to propagate over high-order neighborhoods, generalizing the topology remediation to high-order message passing.

### A.1.2  FRAMEWORK OVERVIEW

We propose a novel GNN architecture to Remedy Topology via Target-oriented homophily (GReTo), which simultaneously performs adaptive immediate neighborhood aggregation and high-order layer propagation, from the perspective of target-oriented homophily disentanglement. Specifically, the proposed two homophily measurements respectively capture node-level intra-graph spatial proximity and inter-graph connections between adjacent steps. Therefore, we take advantage of both two homophily measurements to find target-oriented neighbors, thus calibrating the original topology. Since GNNs have two major processes, i.e., node-wise aggregations and layer-wise propagation, our GReTo on topology-task remediation correspondingly consists of two stages, Singed target-oriented message passing, and Personalized high-order layer propagation, as illustrated in Figure 4. We also detailedly demonstrate the spatial and transition homophily as well as the step-wise topology refinement in Figure 5.

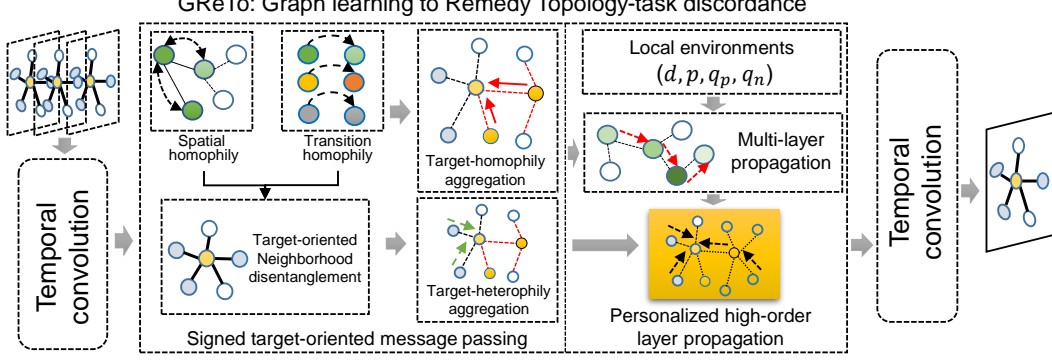

Figure 4: Framework overview of GReTo on remedying topology-task discordance

### A.1.3  TECHNICAL DETAILS OF TRANSITION HOMOPHILY PREDICTOR

**Motivations.** Although we have derived the formation of target-homophily neighborhood, the transition homophily from step $T$ to $T + 1$ cannot be available. However, realizing a predictor to estimate the node-wise transition homophily is intractable due to following two challenges. 1) the unavailability of $x^{T+1}$ and 2) computing transition homophily between all node pairs is computationally inefficient.

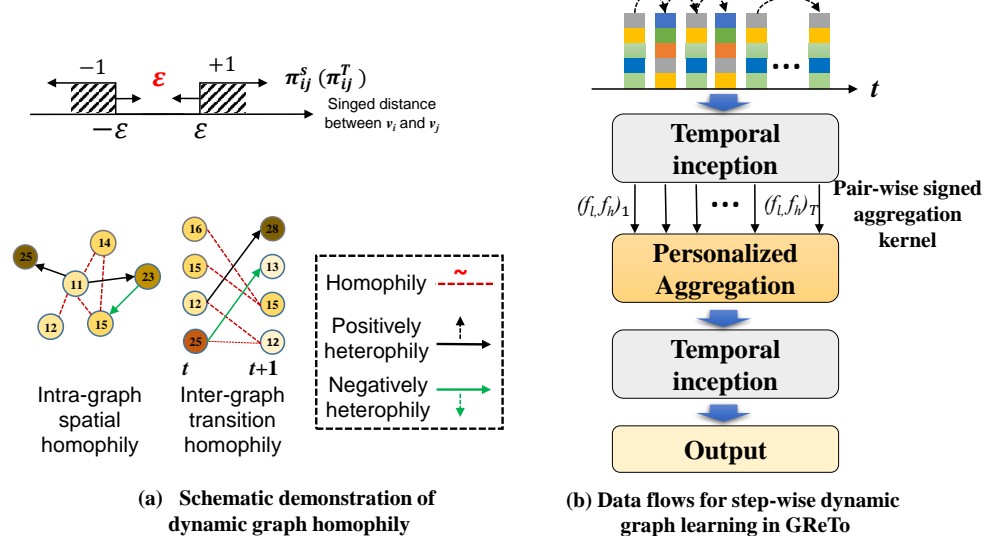

Figure 5: Homophily illustration and step-wise dynamic graph learning

To this end, we make one simplification and convert the homophily estimation into an efficient sequence classification task to address above issues. First, we reduce the transition homophily to a self-transition homophily $Sig_i^t$. Our $Sig_i^t = (\pi_{ii}^T)|_{t-1 \to t}$ measures the degree of node-level self variations between adjacent

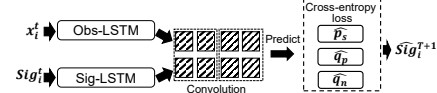

Figure 6: Transition homophily predictor

steps, which can be available from historical observations and encourages efficient computations. Second, to complement the unavailable $x^{T+1}$, we find the serial inter-graph transition homophily obeys the favorable property of time-series, i.e., the periodicity and continuity. Therefore, we formulate a time-series classification task.

**Techniques.** Concretely, the time-series learner concatenates two parallel LSTMs and one 1D-convolution (Conv1D) to capture the evolution and trend dependence by taking all available historical observations of both node values and self-transition homophily $[\boldsymbol{X}_t; \boldsymbol{Sig}^t](t = 1, 2, .., T)$ as inputs. Thus, we obtain the estimated next-step representation by,

$$\widehat{\boldsymbol{Sig}}_R^{T+1} = \text{Conv1D}(\text{LSTM}(\boldsymbol{X}_1, \boldsymbol{X}_2, ..., \boldsymbol{X}_T); \text{LSTM}(\boldsymbol{Sig}^1, \boldsymbol{Sig}^2, ..., \boldsymbol{Sig}^T)) \quad (15)$$

We then exploit the learned time-series representation $\widehat{\boldsymbol{Sig}}_R^{T+1}$ for a three-category classification where the class probability is estimated with an MLP parameterized by $\boldsymbol{W}_s$, i.e.,

$$[\widehat{\boldsymbol{p}}, \ \widehat{\boldsymbol{q}}_p, \ \widehat{\boldsymbol{q}}_n]^{T+1} = \text{MLP}(\widehat{\boldsymbol{Sig}}_R^{T+1}; \boldsymbol{W}_s) \quad (16)$$

The estimated $\widehat{\boldsymbol{p}}, \ \widehat{\boldsymbol{q}}_p, \ \widehat{\boldsymbol{q}}_n$ refer to three probabilities that $\widehat{\boldsymbol{Sig}}_R^{T+1}$ belongs to corresponding categories, and then we can obtain the final $\widehat{\boldsymbol{Sig}}^{T+1}$ based on these probabilities.

### A.1.4 TECHNICAL DETAILS ON TEMPORAL INCEPTION LAYER

Instead of highly computational recurrent learning (Bai et al., 2020; Li et al., 2018), we adopt an efficient fully-convolution based sandwich structure for temporal inception, following the learning order of 'temporal-spatial-temporal' (Yu et al., 2018). Specifically, the temporal convolution kernel $\boldsymbol{\Gamma}$ is implemented as a gate by taking self observations as the feature inputs to control the message passing. By involving both upper and bottom temporal blocks with our GReTo, the whole learning process can be written as,

$$\widetilde{\boldsymbol{X}} = \boldsymbol{\Gamma} *_\kappa \boldsymbol{X} = \boldsymbol{X} \odot \sigma(\boldsymbol{X}); \ \ \boldsymbol{H}^{(K)} = \boldsymbol{F}_g^*(\widetilde{\boldsymbol{X}}) \quad (17)$$

$$\widehat{\boldsymbol{Y}} = \boldsymbol{\Gamma} *_{\kappa} \boldsymbol{H}^{(K)} = \boldsymbol{H}^{(K)} \odot \sigma(\boldsymbol{H}^{(K)}) \tag{18}$$

Then the convolution is implemented by the element-wise Hadamard product $\odot$ between self-observation and the gate, and $\kappa$ is the length of convolution kernel for temporal aggregation following the setting of (Yu et al., 2018)[4]. With this well-designed structure, our solution can not only sufficiently learn the temporal dependence, but also avoid the insufficiency of RNN-based methods.

## A.2 RELATED WORK

In this section, we revisit a series of related works on Graph Neural Networks (GNNs), homophily theory in GNNs as well as dynamic graph learning.

### A.2.1 GRAPH NEURAL NETWORKS (GNNS)

Graph-structured data has received significant interest to energize substantial domains including molecular modeling (Rong et al., 2020; Beaini et al., 2021), social networks (Zhao et al., 2021) and smart grids (Lin et al., 2021). Early GNNs, which categorized to spectral methods, extend the convolution from grid data to irregular graph data by imposing Fourier transformations on graphs and extract informative structures (Defferrard et al., 2016; Xu et al., 2018). After that, researchers find that spectral GNNs suffer the issues of high computational costs and non-alignment to vertex domain. Therefore, spatial methods are proposed to relax the equal number neighbors to flexible neighborhood constructions by directly defining the kernel functions (Hamilton et al., 2017; Kipf & Welling, 2016; Niepert et al., 2016). To enable a global node-wise correlation learning, attention methods of Graph Attention Networks (Veličković et al., 2018) and ASTGNN (Guo et al., 2019) are subsequently proposed. However, graph-structured data in real scenarios are more complex than assumptions in most existing research. For example, multi-type edges and multi-type nodes in graphs are pervasive in many areas such as chemical bonds in molecules and heterogeneous relationships in social networks or citation networks (Wang et al., 2022a). Also, with the increasing layers of deep graph neural networks, node representations tend to go towards similar by iterative aggregations. To address these challenges, heterogeneous GNN (Zhang et al., 2019), adaptive relational graph learning (Wang et al., 2022a; Schlichtkrull et al., 2018), and heterophily GNNs (Yan et al., 2021; Ma et al., 2021) have raised. Even so, adapting the GNN architecture to more complex scenarios still remains as emerging research topic and less explored.

### A.2.2 HOMOPHILY AND HETEROPHILY IN GNNS

The homophily-heterophily theory aims to quantify local neighborhood distributions, i.e., the proportions of different types of neighbors that approaching and deviating from targets. Given the 'opposites attract' phenomenon in real-world graphs, emerging GNNs are proposed to tackle the heterophily issue. (Bo et al., 2021) first explain multi-type node-wise correlations with low and high frequency components and design frequency adaptation GCN to allow both high and low frequency message passing. After that, Yan et, al. propose a signed message passing and degree-correction (Yan et al., 2021), to enable actively identifying different components on graphs by extracting signs of node-level pair-wise similarity, and re-weighting layer-wise information by degree corrections. At the same time, Zhu et, al. design a compatibility matrix to model the node heterophily (Zhu et al., 2021b) and investigate three principles considering neighboring aggregation and high-order propagations, to adapt heterophily scenarios (Zhu et al., 2020). Actually, the homophily ratios of different substructures can experience large variances (Du et al., 2022). In this way, GPR-GNN (Chien et al., 2020) and MixHop (Abu-El-Haija et al., 2019) enable learnable weights over different layers to achieve high-order propagation. Unfortunately, these high-order propagations cannot specify the weights over node-level, and all above solutions are tailored for classification.

### A.2.3 LEARNING ON DYNAMIC GRAPHS

Majority of dynamic graph learning can be classified into link prediction and node-level regressions. Tasks of link predictions unversally exist in recommendation system (Xia et al., 2022; Göpfert et al., 2022), social networks (You et al., 2022) and heterogeneous academic networks (Kang et al., 2022) where these studies exploit the dynamics of user preference and interests to achieve time-varying

---

[4]The size for temporal convolution kernels is set to 1*3 across four datasets.

Table 3: Dataset statistics

| Dataset | Node # | Time step # | Time span | Interval length | Intra-graph homophily |
|---|---|---|---|---|---|
| Metr-LA | 207 | 34,272 | 03/01/2012-06/30/2012 | 5min | 0.2273 |
| PeMS-Bay | 325 | 52,116 | 01/01/2017-05/31/2017 | 5min | 0.1073 |
| KnowAir | 184 | 11,688 | 01/01/2015-12/31/2018 | 3h | 0.2481 |
| Temperature | 184 | 11,688 | 01/01/2015-12/31/2018 | 3h | 0.1156 |

Table 4: Neighborhood distributions on real-world datasets

| | Intra-graph homophily $\boldsymbol{\pi}_{v_i}^s$ | | | Inter-graph homophily $\boldsymbol{\pi}_{v_i}^T$ | | |
|---|---|---|---|---|---|---|
| | $p_s$ | $q_p$ | $q_n$ | $p_s$ | $q_p$ | $q_n$ |
| Metr-LA | 0.2273 | 0.4732 | 0.2995 | 0.3325 | 0.4920 | 0.1755 |
| PeMS-Bay | 0.1073 | 0.5912 | 0.3015 | 0.2399 | 0.6863 | 0.0738 |
| KnowAir | 0.2481 | 0.3945 | 0.3574 | 0.3190 | 0.4030 | 0.2780 |
| Temperature | 0.1156 | 0.6980 | 0.1864 | 0.1418 | 0.5538 | 0.3044 |

embedding. This research line pays more attention to individual embedding rather than the holistic graph. Node-level regressions such as traffic forecasting (Yu et al., 2018; Guo et al., 2019; Miao et al., 2022) and smart grid predictions (Lin et al., 2021; Chen et al., 2021b), aim to aggregate relevant nodes for approaching future observations. To capture the dynamic topology, recent works devise various adaptive topology learning strategies, including dynamic similarity (Yan et al., 2021; Zhou et al., 2020; Bai et al., 2020) and attention mechanism (Guo et al., 2019), to quantify weights of different nodes. Actually, dynamic graph learning tends to be more challenging than static graphs, as these node values are often time-varying and naturally induce the edge diversity over time (Chen et al., 2020; Yan et al., 2021). In addition, the imbalanced local structures over nodes (Chen et al., 2021a; Wu et al., 2021) further pose challenges to high-order propagation and topology refinement. Although there are studies respectively focus on dynamic relation embedding (Deng et al., 2020), and personalized receptive field construction (Xiao et al., 2021), there are two issues have been significantly neglected. 1) None of them take the temporal evolution into account to capture their influences on their potential dynamic topology and 2) Considering the potential topology-task discordance discussed above, none of them investigate the node-level multi-relations over graphs to facilitate regression tasks.

## A.3 ADDITIONAL EXPERIMENTAL DETAILS AND RESULTS

### A.3.1 STATISTICS OF DATASETS

As this work studies the node-wise proximity over dynamic graphs and investigates how the proximity of both homophily and heterophily enhances dynamic graph regressions. We thus present the statistics of node numbers, total temporal steps and intra-graph homophily of each real-world dynamic graph in Table 3. Also, the numerical neighborhood distributions of each node are averaged into $p_s$, $q_p$ and $q_n$ regarding both two types of homophily, i.e., intra- and inter-graph homophily in Table 4. Noted that the reported numerical homophily is derived with the well-set homophily criteria $\varepsilon$, i.e., 0.08, 0.1, 0.1 and 0.12 corresponding to four dynamic graph datasets. Due to the inheret dynamics in these graphs, we can observe that these graphs suffer serious heterophily issue and the inter-graph homophily tends to be higher than that of intra-graph. We especially observe that datasets of KnowAir and Temperature are with higher negative heterophily $q_n$, and experimental results in Table 1 also demonstrate a more significant improvement on KnowAir and Temperature. These statistics can verify the intuition that modeling homophily and heterophily can disentangle the target-oriented homophily and boost both aggregation and representation capacity in GNNs.

### A.3.2 DETAILS OF COMPARED BASELINE MODELS

**Classic GNNs:** (1) GCN: The classical graph convolution neural network (Kipf & Welling, 2016). (2) GAT: An attention-based graph method capturing global dependence on the whole graph (Veličković et al., 2018). (3) GraphSAGE: A representative spatial method of GNNs to realize graph-based inductive learning (Hamilton et al., 2017). **Topology-refined models:** (4) SuperGAT: A self-supervised GAT designed for noisy graphs, which learns expressive attention in distinguishing mislinked neighbors (Kim & Oh, 2020). (5) EGConv: It is a lightweight GCN that designs spatially-varying adaptive filters (Tailor et al., 2021). **Heterophily GNN:** (6) $H_2GCN$: It is a framework for node-level classification on graph with homophily and heterophily components by three effective designs (Zhu et al., 2020). **Spatiotemporal Networks:** (1) STGCN: A graph-based spatiotemporal framework with a sandwich structure of temporal-spatial-temporal learning architecture (Yu et al., 2018). (2) MTGNN: Multivariate time-series forecasting model which is capable of adaptively learning uni-directed relations among variables with a graph learning module (Wu et al., 2020b). (3) GraphWaveNet (GWN): A state-of-the-art graph-based traffic prediction model that integrates TCNs and GCNs (Wu et al., 2019). (4) DCRNN: A diffusion convolutional recurrent neural network, which combines diffusion graph convolutions with RNNs (Li et al., 2018). (5) ASTGNN: An attention-based spatiotemporal network for capturing dynamic ST correlations (Guo et al., 2019).

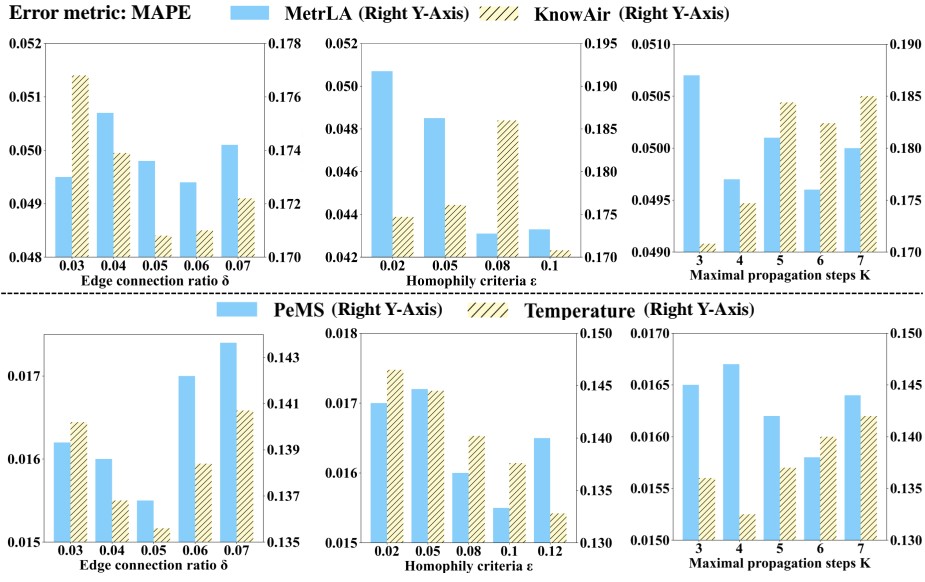

Figure 7: Parameter sensitivity analysis

### A.3.3 ADDITIONAL DETAILED COMPARISON RESULTS

In this subsection, we perform a series of experiments to test the model performance stability with different initializations, model robustness on temporal resolutions and the high temporal node degrees.

Stability comparisons of different models. To explore the performance stability of different models, we train our models with 10 different initialization random seeds, and select three matching baselines, MTGNN, GraphWaveNet (GWN) and DCRNN as the compared baselines. We record the standard deviation of RMSE and MAPE for these models on four datasets and report them in Table 5. Among the four models, our GReTo reveals comparable and identical stability almost on all datasets with other baselines, verifying there is no stability issues in our solution.

**Robustness on temporal resolutions.** Our solution is independent on the length of time intervals as the evolution trend should be captured regardless of the interval length. To confirm this intuition, we perform a case experiment by aggregating the time intervals from 5 min into 30 min on MetrLA

Table 5: Performances with standard deviations

| | Metr-LA (5-min) | | PeMS-Bay (5-min) | | KnowAir | | Temperature | |
|---|---|---|---|---|---|---|---|---|
| | MAPE | RMSE | MAPE | RMSE | MAPE | RMSE | MAPE | RMSE |
| MTGNN | 0.0526±0.0008 | 3.8153±0.0168 | 0.0170±0.0002 | 1.5759±0.0110 | 0.2271±0.0029 | 12.9091±0.0134 | 0.1682±0.0051 | 0.9034±0.0303 |
| GWN | 0.0528±0.0004 | 3.8434±0.0359 | 0.0163±0.0002 | 1.5482±0.0087 | 0.2288±0.0027 | 12.8495±0.0123 | 0.1607±0.0072 | 0.9112±0.0297 |
| DCRNN | 0.0532±0.0006 | 3.8798±0.0459 | **0.0161±0.0003** | 1.6229±0.0092 | 0.2392±0.0024 | 13.0389±0.0317 | 0.1351±0.0057 | 0.9715±0.0803 |
| GReTo | **0.0500±0.0004** | **3.6552±0.0356** | 0.0162±0.00006 | **1.4813±0.0025** | **0.1708±0.0025** | **11.0369±0.0670** | **0.1341±0.0023** | **0.8704±0.0031** |

Table 6: Performances on aggregated 30-min intervals of two datasets

| | Metr-LA (30-min) | | PeMS-Bay (30-min) | |
|---|---|---|---|---|
| Models | MAPE | RMSE | MAPE | RMSE |
| MTGNN | 0.0950±0.0017 | 40.6671±0.6227 | 0.0287±0.0002 | 18.0767±0.0785 |
| GWN | 0.0952±0.0022 | 40.7782±0.5130 | 0.0281±0.0003 | 17.7177±0.0832 |
| DCRNN | 0.0989±0.0010 | 41.5420±0.5594 | 0.0292±0.0003 | 17.8241±0.0863 |
| GReTo | **0.0936±0.0004** | **39.2014±0.3306** | **0.0273±0.0002** | **17.5980±0.0619** |

and PeMS-Bay. The results illustrated in Table 6 show that our solution can also have a 1.6% and 2.8% improvement on these two well-studied datasets when compared with best baseline, where the performance gains are comparable to 5-min setting. Thus, we exactly demonstrate the independence of our solution on temporal resolutions. Noted that the greater magnitude of errors and larger variations of 30-min datasets is probably attributed to the larger magnitude of node observation values on aggregated datasets.

**Robustness on graphs with higher temporal node degrees.** The graphs with higher temporal node degrees are corresponding to the scenarios that nodes in adjacent/different graphs possess more interconnections, i.e., graphs with higher inter-graph homophily. As illustrated in Table 4, this scenario occurs on two datasets, Metr-LA and KnowAir, as both datasets reveal higher inter-graph homophily i.e., larger $p_s$ (0.3325 on Metr-LA and 0.3190 on KnowAir). On their performances, the reduced errors away from best baselines on both datasets, from 0.0526 to 0.0507 on Metr-LA and from 0.2271 to 0.1708 on KnowAir, can imply the robustness of our GReTo on higher temporal degree nodes.

### A.3.4    INTERPRETABLE INTERMEDIATE RESULTS

To present an intuitive understanding on aggregation process of our GReTo, Figure 8 demonstrates a case study of our GReTo on Metr-LA. Some important central nodes associated with their neighborhoods and the intermediate results of layer-wise importance are visualized. It is observed that nodes in graph usually reveal distributed centers and different centers also tend to be connected. For layer-wise importance, it shows that nodes with lower degrees (e.g., Node 7 and 107) tend to propagate farther than nodes with higher degrees (e.g., Node 28 and 78) where the learnable results can verify the intuition of our layer propagation principle and the effectiveness of our GReTo for high-order propagations.

### A.3.5    PARAMETER SENSITIVITY ANALYSIS

We investigate the three hyperparameters in GReTo to test the model parameter sensitivity. We illustrate the optimal parameter searching process on all four datasets in Figure 7. We have the following three observations. First, we must strike a balance on the edge connections between the lower graph connectivity and reduced graph topology, corresponding to lower and higher connectivity of $\delta$ and $\varepsilon$. Second, it can observed that the larger-scale of graphs tend to require a larger propagation order $K$, indicating more neurons can accommodate better fitting capacity towards to adapt more intensive data. Third, although different graphs have their intrinsic property and corrspond to their optimal parameters, we can still find our solution can be stable across different parameters and discover some regularity on parameter tuning.

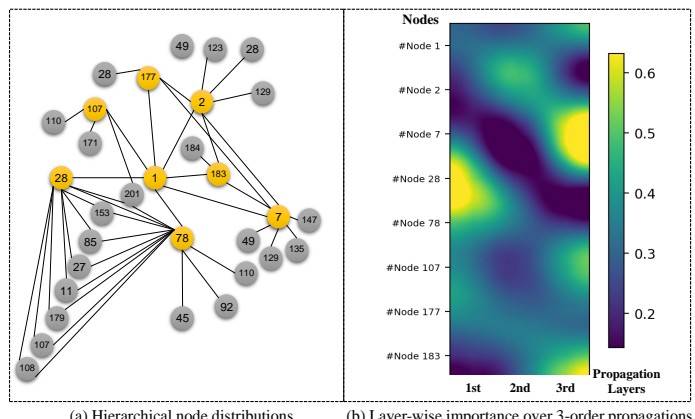

(a) Hierarchical node distributions    (b) Layer-wise importance over 3-order propagations

Figure 8: Case study of GReTo on Metr-LA

Table 7: Model training time for each epoch (unit: seconds)

| Datasets | Metr-LA | PeMS-Bay | KnowAir | Temperature |
|---|---|---|---|---|
| GreTo | 58 | 122 | 16 | 16 |
| GCN | 35 | 87 | 11 | 11 |

### A.3.6 COMPLEXITY ANALYSIS

**Complexity of GReTo.** As model complexity plays an important role in both training and inference stages, we here analyze the additional computation to existing GNNs of our GReTo. For one step $t \in |\mathcal{T}|$, the major computation costs lie in two aspects, (1) in the data-preparation stage, spatial homophily and transition homophily should be pre-calculated, which leads to $\mathcal{O}(N * N)$, (2) in the training and inference stages, real-time spatial homophily computation, a binary judgement for matrix sign disentanglement, additional $K$-times computation for high-order neighborhood distribution as well as matrix product, finally the MLP for LiM, which lead to $\mathcal{O}(2 * K + 1 + 1) = \mathcal{O}(2K + 2)$ times atomic operation (element-wise matrix product or addition). Since the data-preparation stage does not occupy training resources and can be computed only once while the computation during training and inference is linearly increasing with maximal layer number $K$, the additional computation is tolerable for the whole architecture. Besides, we also investigate the training complexity and parameter volumes of the proposed model empirically. The comparison of model training time compared with vanilla GCN is illustrated in Table 7, while the parameter volume comparisons with other comparable dynamic graph learning models are shown in Table 8, where both of them can support the acceptable computations.

**Complexity of GReTo on large-scale graphs.** More specifically, readers may concern about the efficiency issue when the graphs become larger. Here we present the efficiency analysis over very large graphs. In this case, we can force the spatial proximity only to be computed on spatially neighboring nodes. Assuming there are averagely $N_s$ neighbors of each node ($N_s \ll N$), the total computation on a large graph is $N \times N_s$. By adding the computations of temporal evolution, the whole graph homophily calculation only requires totally $N \times (N_s + 1)$ times of computation that is linearly increasing with node numbers, while LIM in the neural network still needs $(2K + 1)$ computation. Further, we can also construct the matrix describing the ego-net neighbors to allow the

Table 8: Parameter volumes on different models (unit: million (M))

| Models | GReTo | MTGNN | GWN | DCRNN |
|---|---|---|---|---|
| Metr-LA | 0.38M | 0.41M | 0.28M | 0.37M |
| PeMS-Bay | 0.38M | 0.57M | 0.37M | 0.28M |

computation to run parallelly. Therefore, with these slight modifications, our solution can be easily generalized to large-scale graphs.

### A.3.7 FURTHER DISCUSSIONS ON EMPIRICAL RESULTS

**Influences of dataset property on module performances.** The ablation studies have illustrated how the model performances vary when each component is individually removed. We find that the improvements induced by designed modules are sensitive to the property of datasets. Since traditional GNNs only aggregate homophily compositions in the current graph for smoothness while our target-oriented message passing explicitly considers the target influences on topology to perform aggregations, the datasets with higher intra-graph heterophily (large $q_n$ of intra-graph) or inter-graph heterophily (large $q_n$ of inter-graph) are more prone to receive more benefits. In fact, more significant performance improvements on KnowAir and Temperature can correspondingly verify our intuition. Therefore, when the property of datasets varies, the performances will change accordingly. Besides, the interactions between two modules may bring in additional bonus where the topology remediation can work cooperatively with the personalized high-order propagation. Concretely, the topology remediation will be meaningless in the high-order neighborhood aggregations if the layer-wise importance measure module is removed, and the ablations on V-ULW (aggregation with uniform layer-wise weight rather than personalized aggregations) exactly confirm this argument.

**Model scalability on different scenarios.** The theoretical analysis and experimental results can verify that our solution is easily generalized to different scenarios. Here, we will analyze the scalability issue on three aspects, i.e., temporal resolution, homophily property and large-scale graph adaptation. First, we argue that our solution is actually independent on the length of time intervals as it only requires to calculate the current graph homophily and captures the potential evolution trend in a data-driven manner. The additional empirical results (Sec. A.3.2) reveal similar improvements on both 5-min and 30-min intervals also demonstrate the independence on the interval length. And for multi-step prediction, as shown in Figure 3 (b), the 6-step prediction results are comparable to other baselines that tailored for multi-step time series predictions (i.e., MTGNN) but with a smaller improvement margin than single step prediction does. This can inspire us to explore an extended version of GReTo-M that is tailored for multi-step prediction. A new GReTo-M can concentrate on how to guide the learning of trends by considering the homophily, context-factors to generalize to a high-quality multi-step prediction. These interesting works will be left as our future work. Therefore, our solution is generally robust to different temporal resolutions. Second, for homophily property, we have illustrated the statistics of neighborhood distribution in Table 4. We find that datasets with different inter-graph homophily and intra-graph homophily exactly reveal various prediction performances. Specifically, our solution explicitly considers the target evolution direction to capture the target-homophily nodes that can potentially involve both heterophily and homophily in one graph. Therefore, our solution can adapt to the high intra-heterophily scenarios such as Temperature, and high inter-graph heterophily scenario such as KnowAir. Actually, the performances on these two datasets have a large margin than baselines that validates our intuitions. Finally, our GReTo can also be easily generalized to large-scale graphs as we are only required to calculate the intra-graph homophily within spatial neighborhoods and inter-graph homophily within node itself. The simplified operations will lead to a linearly increasing computation with node number $N$ that can be satisfactory for model deployment (demonstrated in Sec. A.3.6). To conclude, our newly developed GReTo model is with good scalability on various scenarios probably with slight modifications, and we believe our work can be an advance in both spatiotemporal learning and graph learning theory.

## B PROOF OF THEOREMS

### B.1 PROOF OF THEOREM 1

**Proof.** To prove Theorem 1, we first provide several notations on graph $\mathcal{G}$ and node $v_i$ to facilitate the analysis. (1) $d_i$ serves as the degree of node $v_i$ while the feature value $\mathbf{x}_i$ is bounded by $B$, (2) $\rho(\mathbf{W})$ denotes the largest singular value in learnable parameters $\mathbf{W}$, (3) the ratio of intra-graph spatial homophily $\pi_{ij}^s$ is fixed to $p_i$, while the ratio of other node-level relations is $q_i$, where $q_i = 1 - p_i$, (4) assume a target coefficient $\gamma_i$ quantifies the relationship between the targeted step $T + 1$ and current step $t$ by $x_i^{T+1} = \gamma_i x_i^t$. We omit the non-linear activation layer for simplification, as it will

not intervene this theoretical analysis. Based on above assumptions, we can introduce the following Lemma 1 to demonstrate the representation $h_i$ will finally converge to its expectation.

**Lemma 1** *Acorrding to Hoeffding's Inequality (Bentkus, 2004), the aggregated representation $h_i$ will converge to its expectation $E[h_i]$ with their differences bounded by,*

$$\mathbb{P}(||h_i - E[h_i]||_2 \geq t) \leq 2l \exp(-\frac{d_i t^2}{2\rho^2(\mathbf{W})B^2 F}) \tag{19}$$

Lemma 1 has been exactly proved in literature (Ma et al., 2021) on classification tasks. Considering GNNs share the same process of neighborhood aggregation and node representation updates on both classification and regression tasks, Lemma 1 can be extended to regression tasks.

After then, we can derive the expectation $E[h_i^t]$ of $h_i^t$ to complete following analysis. For simplification, we only consider the evolution from the one step $T$ to the targeted step $T+1$, i.e., $x_i^T \to x_i^{T+1}$, where we assume $x_i^{T+1} = \gamma_i x_i^T$. If only the nodes of intra-graph homophily can be aggregated, each aggregated neighbor $v_j \in \mathcal{N}(v_i)$ satisfies $x_j < (1 \pm \varepsilon)x_i$. Let $x_j = (1 \pm \widetilde{\varepsilon})x_i$, where $\widetilde{\varepsilon}$ satisfy $\widetilde{\varepsilon} < \varepsilon < 1$. Therefore, we can calculate the expectation $E[h_i^T]$ after once convolution, i.e.,

$$E[h_i^T] = \frac{x_i + d_i p_i (1 \pm \widetilde{\varepsilon})x_i}{d_i + 1} \tag{20}$$

where $d_i p_i$ interprets the expectation number of the spatial homophily nodes in the whole graph.

Given the target coefficient $\gamma_i$, a successful aggregation can be equivalent to finding a valid $d_i$ to construct an appropriate neighborhood field for achieving targets. Therefore, we can continue to derive the relationship between $d_i, \gamma_i$, and $p_i$.

The difference between $E[h_i^T]$ and the targeted value $x_i^{T+1} = \gamma_i x_i^T$ can be written as,

$$||E[h_i^T] - x_i^{T+1}|| = ||\frac{1 + d_i p_i (1 \pm \widetilde{\varepsilon}) - \gamma_i(d_i + 1)}{d_i + 1}x_i|| \tag{21}$$

Consider that the differences will converge to 0 as the optimized points are obtained, we can derive the closed solution of $d_i$,

$$d_i = \frac{\gamma_i - 1}{p_i(1 \pm \widetilde{\varepsilon}) - \gamma_i} \approx \frac{\gamma_i - 1}{p_i - \gamma_i} = \frac{1 - \gamma_i}{\gamma_i - p_i} = -1 + \frac{1 - p_i}{\gamma_i - p_i}$$

By analyzing the above inverse proportional function in Figure 9, we achieve the conditional relationship among the substructure $d_i$, the target coefficient $\gamma_i$ and intra-graph homophily $p_i$,

$$d_i = \begin{cases} (-\infty, -1), & \gamma_i < p_i \\ \infty, & \gamma_i = p_i \\ [0, +\infty), & p_i < \gamma_i \leq 1 \\ (-1, 0), & \gamma_i > 1 \end{cases}$$

Based on above analysis, we have theoretically demonstrated that we cannot obtain the optimized $d_i$ when $\gamma_i < p_i$ and $\gamma_i > 1$, corresponding to the discordance between topology and tasks. □

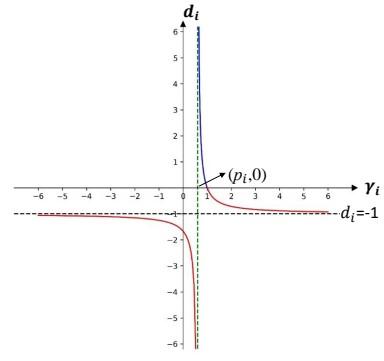

Figure 9: Relationship between the target coefficient $\gamma_i$ and substructure degree $d_i$.

### B.2 PROOF OF THEOREM 2

**Proof.** To faciliate the analysis on more complex message passing, we introduce the following notations and settings. Given node $v_i$, the spatial homophily still keeps as $p_i$ while heterophily ratio keeps as $q_i$, $d_i q_i$ is the expectation number of heterophily neighbors to $v_i$, and $\alpha$ adjusts the aggregation proportions between homophily and heterophily.

We consider the expectation values of positive and negative heterophily neighbors respectively as $\frac{x_i}{\lambda_1}$ and $\lambda_2 x_i$ where $(\lambda_1, \lambda_2)$ satisfying $0 < \lambda_1, \lambda_2 < 1$ are the scalar factors. By formulating a seperated aggregation strategy, we can derive the difference between the expectation of current-graph representation and target as,

$$||E[h_i^T] - x_i^{T+1}||=||\frac{x_i + d_i p_i (1 \pm \varepsilon) x_i + d_i q_i (\frac{\alpha}{\lambda_1} s_p x_i + (1-\alpha)\lambda_2 s_q x_i)}{d_i + 1}$$
$$- \frac{\gamma_i x_i (d_i + 1)}{d_i + 1}|| \tag{22}$$

with

$$\begin{cases} d_i \in \mathbb{Z}^+ \\ 0 < q_i < 1 \\ 0 < \lambda_1, \lambda_2 < 1 \end{cases}$$

Let the difference be 0, we have the following equation,

$$1 + d_i p_i (1 \pm \varepsilon) + d_i q_i (\frac{\alpha}{\lambda_1} - (1-\alpha)\lambda_2) - \gamma_i (d_i + 1) = 0 \tag{23}$$

Ignoring the relatively small $\varepsilon$, we can arrive the optimzed closed form of $d_i$,

$$d_i = \frac{\gamma_i - 1}{p_i + q_i(\frac{\alpha}{\lambda_1} - (1-\alpha)\lambda_2) - \gamma_i}$$

To find an optimized $d_i \in \mathbb{N}^+$, the parameter tuple $(p_i, q_i, \alpha, \gamma_i, \lambda_1, \lambda_2)$ must conform to the following inequality by simplifying $s_p = 1, s_q = -1$,

$$0 < p_i + q_i(\frac{\alpha}{\lambda_1} - (1-\alpha)\lambda_2) - \gamma_i < \gamma_i - 1 \tag{24}$$

Without loss of generality, we denote $\tau^* = \frac{\alpha}{\lambda_1} - (1-\alpha)\lambda_2$ and substitute $q_i$ with $1 - p_i$, then we can simplify Equation 23 into,

$$\gamma_i < p_i + (1 - p_i)\tau^* < 2\gamma_i - 1 \tag{25}$$

Then we can derive the condition that $\tau^*$ must satisfy as,

$$\frac{\gamma_i - p_i}{1 - p_i} < \tau^* < \frac{2\gamma_i - p_i - 1}{1 - p_i} \tag{26}$$

Since the node-wise aggregation process are parameterized by both signed kernels and learnable parameters, it can be viewed as encapsulating learnable $\mathbf{W}$ into our pre-defined parameters $(\lambda_1, \lambda_2, \alpha)$ and hence we can always find the optimal $\tau^*$ to satisfy above inequality. We will further present a detailed analysis of above inequality for verifying the existence of such solution.

Let $\tau^* = t - \frac{1}{t}$, we have,

$$\frac{\gamma_i - p_i}{1 - p_i} < t - \frac{1}{t} < \frac{2\gamma_i - p_i - 1}{1 - p_i} \tag{27}$$

Then we can resolve the inequality and derive the exact $t$ by,

$$\frac{\frac{\gamma_i - p_i}{1 - p_i} + \sqrt{\left(\frac{\gamma_i - p_i}{1 - p_i}\right)^2 + 4}}{2} < t < \frac{\frac{2\gamma_i - (p_i + 1)}{1 - p_i} + \sqrt{\left(\frac{2\gamma_i - p_i - 1}{1 - p_i}\right)^2 + 4}}{2} \tag{28}$$

Ultimately, a set of possible solution $(\lambda_1, \lambda_2, \alpha)$ to the directional message passing should satisfy the following inequality,

$$\frac{\frac{\gamma_i - p_i}{1 - p_i} + \sqrt{\left(\frac{\gamma_i - p_i}{1 - p_i}\right)^2 + 4}}{2} < \frac{\alpha}{\lambda_1} = \frac{1}{(1-\alpha)\lambda_2} < \frac{\frac{2\gamma_i - (p_i + 1)}{1 - p_i} + \sqrt{\left(\frac{2\gamma_i - p_i - 1}{1 - p_i}\right)^2 + 4}}{2} \tag{29}$$

Given that $\lambda_1$ and $\lambda_2$ respectively imply the neighborhood of intra-graph positively heterophily and negative heterophily, in the following section, we will continue to analyze what conditions will $\lambda$ satisfy when $\gamma_i$ tends to approach respectively 0 or $+\infty$. This will provide insights into the evaluations of predictability and GNN success on the given datasets.

1) when $\gamma_i \to 0$, we have,

$$\lim_{\gamma_i \to 0} d_i = \frac{\gamma_i - 1}{p_i + q_i(\frac{\alpha}{\lambda_1} - (1-\alpha)\lambda_2) - \gamma_i} \tag{30}$$

To ensure the a valid $d_i$ with $1 \leq d_i \leq +\infty$, we derive,

$$\gamma_i - 1 < p_i + q_i(\frac{\alpha}{\lambda_1} - (1-\alpha)\lambda_2) < \gamma_i \tag{31}$$

We can obtain the constraints of $\lambda_2$,

$$\frac{\alpha}{\lambda_1(1-\alpha)} < \lambda_2 < \frac{1}{\lambda_1} + \frac{p_i - \gamma_i}{(1-\alpha)q_i} \tag{32}$$

Thus, for $\gamma_i \to 0$, the successful aggregation can be achieved when compositions of negatively heterophily neighbors dominate the neighborhoods, i.e., we have a large $\lambda_1$ but small $\lambda_2$ satisfying Eq. 31.

2) when $\gamma_i \to +\infty$, we have,

$$\lim_{\gamma_i \to +\infty} d_i = \frac{\gamma_i - 1}{p_i + q_i(\frac{\alpha}{\lambda_1} - (1-\alpha)\lambda_2) - \gamma_i} \tag{33}$$

To ensure the $1 \leq d_i \leq +\infty$, we derive,

$$0 \leq p_i + q_i(\frac{\alpha}{\lambda_1} - (1-\alpha)\lambda_2) - \gamma_i \leq \gamma_i \tag{34}$$

As the assumption of $\gamma_i \to +\infty$ but $0 < \alpha, \lambda_2, p_i, q_i < 1$, we can directly neglect the components of both $p_i$ and $(1-\alpha)\lambda_2 q_i$. We then have,

$$\begin{cases} \frac{q_i\alpha}{2\gamma_i - p_i + q_i(1-\alpha)\lambda_2} < \lambda_1 \lessgtr \frac{q_i\alpha}{\gamma_i - p_i + q_i(1-\alpha)\lambda_2} \\ \lambda_2 > \frac{\alpha}{(1-\alpha)\lambda_1} \end{cases} \tag{35}$$

Thus, for $\gamma_i \to +\infty$, the successful aggregation can be achieved when compositions of positively heterophily neighbors dominate the neighborhoods, i.e., we have a small $\lambda_1$ but large $\lambda_2$ satisfying Eq. 34.

Therefore, we conclude that extending the aggregation to signed message passing with separately aggregating different types of neighborhoods can exactly enhance the aggregation capacity. □

**Remark.** In Theorem 2, we demonstrate the required conditions of expected neighbors that potentially lead to successful aggregations under signed message passing. With these derived conditions, we can roughly estimate whether the GNN can success or not to aggregate the neighborhoods for achieving targets if the dataset is given. Based on Theorem 2, we can remedy the task-topology discordance by disentangling the neighborhood relationships and performing signed bi-directional message passing.

### B.3 PROOF OF THEOREM 3

**Intuitive demonstration.** As illustrated in Figure 10, we can observe that non-identical neighborhood environments can directly result in different optimized aggregation steps (orders) on each neighboring direction. In other words, the neighborhood topology and the target-heterophily nodes determine the stopping step of aggregation along each route. We demonstrate the detailed proof regarding their quantitative relationship as below.

**Proof.** Due to the mixed compositions of intra-graph homophily and heterophily in one graph, there must be target-heterophily edges. Intuitively, the target-oriented neighboring nodes, i.e., target-homophily are more beneficial to the aggregation process leading to targets. With the notations and basic setting in Theorem 3, we can analyze the expected number of total target-homophily neighbors. To simplify the proof, we leverage local dependence assumption in Graph Information Bottleneck (GIB) theory (Wu et al., 2020a) in our analysis, which assumes nodes in a same $k$-order EgoNet share the same neighborhood distributions and environments. Also, herein we only consider a two-layer propagation ($k$=2) and let the two-order neighborhood share the same **local neighborhood**

**environments** $(d_i, p, q_p, q_n)$ specified by a central node $v_i$ (we will not impose the node index subscript to $(p, q_p, q_n)$ as we make the local dependence assumption in the local fields).

We first consider the scenairo of a large $\gamma_i$ in a consecutive transition $x_i^{T+1} = \gamma_i x_i^T > x_i^T$ $(\gamma_i > 1)$, and there are two cases that nodes are beneficial to aggregate. The beneficial nodes are distributed hierarchically and carefully depicted in Figure 11(a), i.e., (1) in the first-order propagation, the positively intra-graph heterophily neighbors to central node (circle colored in green), (2) in the second-order propagation, the positively intra-graph heterophily compositions neighboring to first-order homophily neighbors (left ellipse colored in blue), and the intra-graph homophily and positive heterophily compositions neighboring to first-order neighbors that are positively heterophily compositions to $v_i$ (right ellipse colored in blue). Let $N_+(v_i)$ be the expected number of target-homophily neighbors of $v_i$, we can derive the following equation,

$$
\begin{aligned}
N_+(v_i) &= q_p d_i + pq_p d_i^2 + (pq_p + q_p^2)d_i^2 \\
&= d_i^2 q_p^2 + q_p(2pd_i^2 + d_i)
\end{aligned}
\tag{36}
$$

Denoting the expected feature values of first-order target-oriented neighborhood as $x_{o1}$ and feature observations of the second-order neighborhood as $x_{o2}$, the expected aggregation observations normalized by the degree $d_i$ can be derived as,

$$
\begin{aligned}
E[h_i^2] &= \frac{1}{d_i} \times q_p d_i \times x_{o1} + \frac{1}{d_i^2}(2pq_p + q_p^2)d_i^2 x_{o2} \\
&= q_p x_{o1} + (2pq_p + q_p^2)x_{o2}
\end{aligned}
\tag{37}
$$

Therefore, we consider the information aggregated from target-oriented nodes as the probability that beneficial nodes are exactly selected for aggregation. In this way, the difference of such probability between the second layer and the first layer $\Delta P(N_+)|_{1-2}$ is computed by,

$$
\begin{aligned}
\Delta P(N_+)|_{1-2} &= P(N_+)_2 - P(N_+)_1 \\
&= 2pq_p + q_p^2 - q_p \\
&= 2(1 - q_p - q_n)q_p + q_p^2 - q_p \\
&= (-1 - 2\eta)q_p^2 + q_p
\end{aligned}
\tag{38}
$$

We consider the quadratic function of $q_p$, where the function describes the compositions of target-homophily.

1) If $0 < q_p < \frac{1}{1+2\eta}$, we have $\Delta P(N_+) > 0$. It refers to that when it is with a small target-homophily, the second-layer can gain more information than the first-layer does, i.e., we have $INFO(h_i^2) > INFO(h_i^1)$. Similar conclusion can be extended to $INFO(h_i^4) > INFO(h_i^3)$ for 3-rd and 4-th layer propagation, and further higher-order propagations if they share similar neighborhood environments. This numerical result can support the intuitive understanding that the smaller target-homophily is, the larger layer propagation steps are required to gain more informative target-related messages.

2) If $q_p > \frac{1}{1+2\eta}$, we have $\Delta P(N_+) < 0$. This interprets that a large target-homophily can encourage the first-layer to gain more information than the second one, i.e., we have $INFO(h_i^1) > INFO(h_i^2)$, and similarly $INFO(h_i^3) > INFO(h_i^4)$ achieves. An intuitive understanding can be achieved that the larger target-homophily, the fewer propagation steps are demanded to balance the propagation efficiency and information gains.

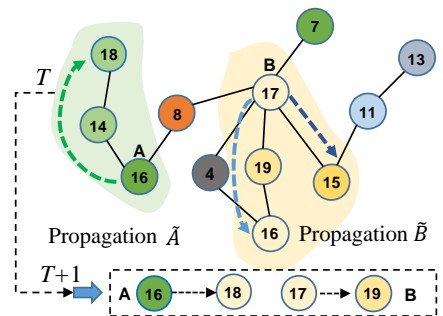

Figure 10: Illustration of node-specific personalized receptive field

For the scenario of small $\gamma$, the negatively intra-graph heterophily compositions are encouraged to aggregate, as illustrated in Figure 11(b). Now the proportion of target-oriented composition becomes $q_n$ in the first-order propagation. Let $N_-(v_i)$ be the expected number of target-homophily neighbors of $v_i$, we have,

$$
N_-(v_i) = q_n d + pq_n d^2 + (q_n d(p + q_n)d)
\tag{39}
$$

Similar to above analysis, the probability difference of aggregating the nodes that are target-oriented (intra-graph negatively heterophily) between the second and first layers can be derived as,

$$\Delta P(N_-)|_{1-2} = (-1 - 2\eta)q_n^2 + q_n \tag{40}$$

Eq. 39 is highly analogous to Eq. 37, hence the symmetric results can be achieved when we have a small $\gamma_i$ satisfying $\gamma_i < 1$.

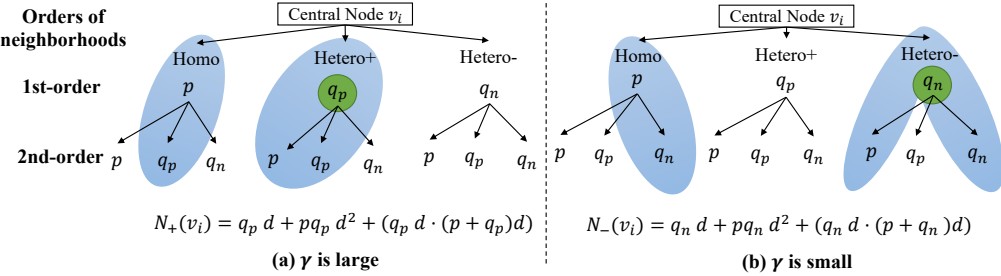

Figure 11: Example of target-homophily node selection process. The highlighted circles in green represent the target-oriented neighborhood in first propagation order while the circles in blue indicate the target-oriented neighborhood in the second order. Homo, Hetero+ and Hetero- respectively refer to target-homophily, postively target-heterophily and negatively target-heterophily.

We can finally complete the proof of the quantitative relationship between the informativeness of layer-wise node representation and the target-oriented homophily ratio. □

### B.4 BROADER IMPACTS AND ETHICS ISSUE

**Scalability and broader impacts.** In general, this paper proposes a novel GNN architecture, GReTo, for dynamic graph regressions via remedying the topology-task discordance. Our work provides a new perspective that exploits both original topology and targets to investigate topology refinement and reconstruction, i.e., the topology in structured data must be an informative knowledge for message passing while the aggregations may not involve all connected nodes but only the target-beneficial neighborhoods. This research delves into how fine-grained analysis of neighborhood compositions facilitates dynamic graph regressions and can promisingly inspire better modeling and analysis on various graph-structured data with complex explicit or implicit correlations. In detail, it not only can be a good baseline for multivariate series prediction such as smart grid prediction, numerical weather forecasting, and economic growths, but it can also be generalized to resolve various classification tasks on dynamic graphs including link predictions in social networks, citation networks, and recommendation systems.

**Limitations and future works.** The limitation of our work can be summarized as 1) limited direction expressivity for multi-step predictions, 2) lacking individual-level node analysis and interpretability on controllable message passing, i.e., dissecting how each composition of neighborhood affect the representation, deviating or approaching targets? Therefore, for future works, we are going to comprehensively study 1) how to exploit homophily theory to improve more general dynamic graph learning tasks such as link predictions or edge-type predictions, 2) how the node-level relations and neighborhood compositions affect the node representations, which potentially can control the message passing in a robust and anti-noise manner.

**Fairness and ethic issues.** Our work performs extensive analysis and experiments on datasets including traffics on two cities and climate datasets concerning air quality and temperature, without any personal identity and privacy issues. Therefore, our work is with no ethics and privacy issues. In addition, all baselines and methods are compared with fairness.

