# OpenReview forum: "GReTo: Remedying dynamic graph topology-task discordance via target homophily"
_ICLR.cc/2023/Conference — ICLR 2023 poster_

### Official Review · Reviewer_1rTB · 2022-10-23

**Confidence:** 4
**Correctness:** 3
**Technical Novelty And Significance:** 3
**Empirical Novelty And Significance:** 3
**Recommendation:** 6

**Clarity, Quality, Novelty And Reproducibility:**

This paper is well-written and easy to follow. For novelty, the authors propose a a novel GNN to remedy topology-task discordance on dynamic graphs. For reproducibility, the authors should provide source code and cross-domain dynamic graph datasets to help better understand their work.

**Strength And Weaknesses:**

Strength:

1.The paper is well organized and clearly written.

2.The theoretical analysis is solid.

3.The framework of GReTo is intuitively, where the two parts are well designed and well described.

Weaknesses:

1.Although dynamic graph regression shows positive results, there should be more insightful analysis to show where the improvements come from. In addition, mean and standard deviation for each method are recommended to make the experimental results more convincing.

2.The ablation result analysis is not very convincing. Except the Temperature dataset, the variant V-Mix is always close to the full variant version, which means that not always all the components are needed.

3.It is necessary to discuss whether the proposed GReTo can learn on dynamic graphs with large average time intervals and higher temporal node degrees.

4.Some minor mistakes, Figures 2 and 7 are repeated.

**Summary Of The Paper:**

This manuscript presents a novel GNN, namely GReTo, to remedy topology-task discordance on dynamic graphs via target-homophily. They first formalize a dynamic graph homophily theory with both signs and distances, and accordingly refine the topology by designing the signed target oriented message passing and the layer importance based propagation. Experimental results on four dynamic graphs verify the superiority of proposed GReTo.

**Summary Of The Review:**

The authors propose a novel GNN to remedy the topology-task discordance on dynamic graphs by integrating the signed message and layer-importance, which is reasonable and solid. The paper is well written, and easy to follow. However, more experimental details and analysis are recommended to be considered and provided.

---

> ### Author Response · Authors · 2022-11-06
> **Response to Reviewer 1rTB (1/2)**
>
> Dear Reviewer 1rTB,
>
> &emsp;We are very grateful to have your positive feedback. Many thanks! We will carefully address your concerns as below.
>
> **Comment 1:** Although dynamic graph regression shows positive results, there should be more insightful analysis to show where the improvements come from. In addition, mean and standard deviation for each method are recommended to make the experimental results more convincing.
>
> **Response to 'further analysis of where improvements come from':** We will provide the responses on three aspects.
>
> 1. **Improvements illustrated by ablation studies.** The ablation studies have illustrated how the model performance will vary when each component is individually removed. In detail, the target-oriented design of our GReTo can largely benefit the learning of almost all datasets, which identifies target-homophily and target-heterophily neighbors and aggregates them separately. Besides, the improvements are also sensitive to the property of datasets. Since traditional GNNs only aggregate homophily compositions in the current graph for smoothness while our work explicitly considers the heterophily and remedies the topology from the perspective of task orientation, i.e., explicitly considering the target influences on topology, the datasets with higher intra-graph homophily (large $q_n$ of intra-graph) or inter-graph homophily (large $q_n$ of inter-graph) are more prone to receive more benefits. In fact, learning on KnowAir and Temperature show more significant promotions correspondingly and verify our intuition. For different strategies of high-order (multi-gnn layer) propagation, these both promising results can demonstrate the necessity of high-order propagation and furthermore, our fine-grained masking mechanism can be more robust on all datasets.
>
> 2. **Improvements illustrated by case studies.** The detailed case study and model behavior analysis can be found in Figure 9. The case intuitively verifies the intuition that the nodes with lower topology degrees is expected to propagate farther with larger layer importance in their subsequent propagation steps.
>
> 3. **Improvements illustrated by theoretical analysis.** Theorem 2 proves that the improved aggregation capacity can be obtained by extending to the sign-preserved aggregation scenario with the derivation of the solution existence of node degree. And Theorem 3 provides the informativeness analysis on how deep each node should propagate and guide the propagation principle.
> To summary, the first aspect demonstrates the improvement from target-oriented design while the latter two aspects show the layer-wise importance measure can lead to promotions. We hope above analysis can facilitate the understanding of our performance improvements.
>
> **Response to 'providing results with standard deviation':** Since we have tested the parameter sensitivity on all datasets and we find that our solution is not very sensitive to hyperparameters, we have not provided the standard deviation but only provide the averaged performance with 5-round tests. We are conducting the experiments day and night to add the deviations to guarantee the completeness of our work. More updates of numerical experiments will follow.
>
> **Comment 2:** The ablation result analysis is not very convincing. Except the Temperature dataset, the variant V-Mix is always close to the full variant version, which means that not always all the components are needed.
>
> **Response:** Thank you for your comments. But we believe that the empirical results cannot fully (absolutely) account for the technical contributions. Our reasons are three-fold.
>
> 1. **The experimental results are relied on the property of datasets.** Different modules are with diverse sensitivity to datasets with different properties, where the property can be the size of graphs, the degree of inter-graph homophily and intra-graph homophily, etc. As discussed in **Comment 1**, with higher inter-graph heterophily, our GReTo can contribute more to the improved performances where we actively identify the target-oriented neighbors regardless of the original homophily degree on current graphs. Therefore, when the property of datasets varies, the performances will change accordingly and we may not attribute these variations on the module designs.
>
> 2. **The interactions between two modules.** The interactions between two modules can bring in additional bonus where the topology remediation should cooperatively work with the personalized high-order propagation. Otherwise, if the personalized high-order propagation is removed, the topology remediation will be meaningless in the high-order neighborhood aggregations. In addition, the empirical improvements across four datasets, even slight on several of them, can still provide the evidence of effectiveness of our modules.
>
> 3. **Theoretical analysis.** We also provide the theoretical analysis of Theorem 3 to illustrate our technical contributions.

---

> ### Author Response · Authors · 2022-11-06
> **Response to Reviewer 1rTB (2/2)**
>
> **Comment 3:** It is necessary to discuss whether the proposed GReTo can learn on dynamic graphs with large average time intervals and higher temporal node degrees.
>
> **Response:**  Our solution is independent on the length of time intervals (temporal resolution) as the evolution trend will always be captured regardless of the interval length. For higher temporal node degrees, we guess you may refer to the scenarios with higher inter-graph homophily where nodes in adjacent/different graphs tend to possess with more connections. Illustrated in Table 3, the two datasets, this scenario occurs on Metr-LA and KnowAi, as both datasets reveal higher inter-graph homophily i.e., larger $p_s$ (0.3325 on Metr-LA and 0.3190 on KnowAir). The reduced errors away from best baselines on both datasets Metr-LA (from 0.0526 to 0.0507) and KnowAir (from 0.2271 to 0.1708) can demonstrate the robustness on higher temporal degree nodes.
>
> To confirm this intuition of the independece on temporal resolution, we will additionally perform case experiments by aggregating the time intervals from 5 min into 30 min on both datasets of Metr-LA and PeMS-Bay. We will provide the results here upon these experiments are completed.
>
> -----------------------------Update---------------------------------
>
> We have selected three comparable baselines, MTGNN, GWN, DCRNN, along with our GReTo and run them on the aggregated 30-min datasets of Metr-LA and PeMS-Bay. By working day and night, the results have been obtained and are reported as below. Noted that we have run them for 10 times and recorded the standard deviation for each method.
>
> Permances on 30-min Metr-LA:
>
> |       | MAPE          | RMSE            |
> | ----- | ------------- | --------------- |
> | MTGNN | 0.0950±0.0017 | 40.6671± 0.6227 |
> | GWN   | 0.0952±0.0022 | 40.7782± 0.5130 |
> | DCRNN | 0.0989±0.0010 | 41.5420± 0.5594 |
> | GReTo | 0.0936±0.0004 | 39.2014±0.3306|
>
>
> Permances on 30-min PeMS-Bay:
>
> |       | MAPE          | RMSE           |
> | ----- | ------------- | -------------- |
> | MTGNN | 0.0287±0.0002 | 18.0767±0.0785 |
> | GWN   | 0.0281±0.0003 | 17.7177±0.0832 |
> | DCRNN | 0.0292±0.0003 | 17.8241±0.0863 |
> | GReTo | 0.0273±0.0002 | 17.5980±0.0619 |
>
>
>
> As illustrated, our GReTo can still reveal improvements when compared with best baselines, respectively obtaining 1.6\% and 2.8\% improvements on these well-studied two datasets. Thus, our solution is independent on the temporal resolution of the dataset. Noted that the greater magnitude of errors and larger variations of 30-min datasets is probably attributed to the larger magnitude of node observation values on aggregated datasets.  We will incorporate these results and additional discussions in our revised submission. Thank you for your kind advice!
>
> **Comment 4:** Some minor mistakes, Figures 2 and 7 are repeated.
>
> **Response:** Figure 7 is illustrated by coordinating with the discussions of motivation of our transition homophily predictor. Thus, it may not be a mistake. Thanks all the same.
>
> **Comment 5:** Authors should provide source code and cross-domain dynamic graph datasets to help better understand their work.
>
> **Response:** Actually, we have provided the source code in the supplementary materials and properly cited the datasets we have employed from previous published literature. We will also publish the processed datasets when accepted. Here we list the refereed papers to illustrate where these datasets come from.
>
> [8] Diffusion convolutional recurrent neural network: data-driven traffic forecasting, ICLR 2018. (Metr-LA and PeMS-Bay)
>
> [9] Pm2. 5-gnn: A domain knowledge enhanced graph neural network for pm2. 5 forecasting, ICAGIS 2020. (KnowAir and Temperature)
>
> We would like to thank Reviewer 1rTB again for the positive feedback and valuable time on our work. We will continue to make efforts to improve our manuscript by supplementing more experimental results and more detailed analysis regarding the results.

---

> ### Author Response · Authors · 2022-11-08
> **Additional experimental reports on model stability**
>
> --------------------------Update------------------------------
>
> Following your suggestions, we continue to work on our experiments to test the performance stability of different methods. Specifically, we train our models with 10 different initialization random seeds, and select three matching baselines, MTGNN, GraphWaveNet (GWN) and DCRNN as the compared baselines. We record the standard deviation of RMSE and MAPE for these models on four datasets and report them in the following Table.
>
> |       |   Metr-LA      |        Metr-LA                 |     PeMS-Bay      |       PeMS-Bay                     |         KnowAir          |    KnowAir    |     Temperature        |  Temperature  |
> |-------|:-------------------------:|:-------------------------:|:--------------------------:|:-------------------------:|:-------------------------:|:--------------------------:|:-------------------------:|:-------------------------:|
> |       |            MAPE           |            RMSE           |            MAPE            |            RMSE           |            MAPE           |            RMSE            |            MAPE           |            RMSE           |
> | MTGNN | 0.0526±0.0008 | 3.8153±0.0168 |        0.0170±0.0002       |      1.5759±0.0110      | 0.2271±0.0029 |       12.9091±0.0134       |       0.1682±0.0051       | 0.9034±0.0303 |
> | GWN   |       0.0528±0.0004       |       3.8434±0.0359       |        0.0163±0.0002       | 1.5482±0.0087 |       0.2288±0.0027       | 12.8495±0.0123 |       0.1607±0.0072       |       0.9112±0.0297       |
> | DCRNN |       0.0532±0.0006       |       3.8798±0.0459       |   0.0161±0.0003  |       1.6229±0.0092       |       0.2392±0.0024       |       13.0389±0.0317       | 0.1351±0.0057 |       0.9715±0.0803       |
> | GReTo |   0.0500±0.0004  |   3.6552±0.0356  | 0.0162±0.00006 |   1.4813±0.0025  |   0.1708±0.0025  |   11.0369±0.0670  |   0.1341±0.0023  |   0.8704±0.0031  |
>
> Among the four models, our GReTo reveals comparable and identical stability almost on all datasets with other baselines, therefore, our solution has no stability issues.
>
> Following your suggestions on reporting standard deviations and more insightful experimental analysis, we will carefully incorporate these additional experiments into our submission, and add a new subsection of “Further discussions on empirical results” that will discuss (1) the influences of dataset property on module performances, and (2) model scalability on different scenarios (performance on different temporal resolutions, homophily property and large-scale graph adaptation). Thanks for your valuable suggestions!
>
> If you have any further questions or concerns on our work, please feel free to raise them.

---

### Official Review · Reviewer_XKsq · 2022-10-30

**Confidence:** 4
**Correctness:** 3
**Technical Novelty And Significance:** 3
**Empirical Novelty And Significance:** 2
**Recommendation:** 8

**Clarity, Quality, Novelty And Reproducibility:**

Clarity: The work is presented clearly but with certain illustrations in the Appendix which requires the reader to jump back and force for better understanding of the proposed techniques.
Quality: The work is of high quality with clear writing and illustrations.
Novelty: Studying of homophily/heterophily (which are known concepts) in the dynamic network setting has marginal novelty.
Reproducibility: Good-- the framework is somewhat complicated but implementations are provided.

**Strength And Weaknesses:**

S1: Novel angle of task-topology discordance for modeling dynamic networks.
S2: Organized presentations with rich theoretical analysis for the proposed techniques.
S3: Promising experimental results on real-world datasets of dynamic networks.

W1: A good motivating example of dynamic task-topology discordance is missing-- we know the concepts of homophily and heterophily in static networks already and it seems natural to consider them in dynamic networks. But in what cases do the homophily and heterophily change?
W2: A schematic figure is missing to clearly visualize different components and their relations in the proposed framework (saw some in the appendix but it may be better to put one in the main paper).
W3: Some case studies to qualitatively understand the behaviors of the models are missing (again saw some in the appendix but it may be better to put some out-- more technical details can be put into appendix to trade for space).


**Summary Of The Paper:**

The paper introduces a dynamic graph neural network model that handles the dynamics in graph homophily wrt specific prediction tasks. Specific modules are designed to model the topology-task discordance along time with theoretical analysis. Experiments are done on real-world traffic and climate datasets.

**Summary Of The Review:**

In general a good paper with a novel angle, solid technical developments, and promising results. Some rearrangements between the main content and appendix may help the presentation.

---

> ### Author Response · Authors · 2022-11-06
> **Response to Reviewer XKsq**
>
> Dear Reviewer XKsq,
>
> &emsp; We are fairly grateful to receive your positive comments. We will elaborately address your concerns by answering the questions and carefully revising our manuscript in the revised version.
>
> **Comment 1:** A good motivating example of dynamic task-topology discordance. But in what cases do the homophily and heterophily change?
>
> **Response:** The differences between static and dynamic networks are lied in two aspects.
>
> 1. The homophily relationship in static graphs will not change over time while node-wise homophily tends to be time-varying with dynamic node values.
> 2. The prediction tasks of static graphs only exploit the information in a same graph while prediction of dynamic graph regression should consider the future step thus the consideration of dynamic homophily and heterophily should incorporate the node-wise relations between temporally adjacent graphs, i.e., inter-graph homophily is required.
>
> Therefore, the homophily relations on graphs will change when the node values change with time. Our work takes the temporal evolution  direction, i.e., target in dynamic graph regressions, into account and performs dynamic message passing by deriving the spatial (intra-graph) homophily and predicting temporal (inter-graph) homophily. Actually, subfigures (a) in Figure 1 discusses this scenario to motivate the dynamic homophily construction. We will add more words to emphasize the dynamics that influence homophily changing. Thanks for your suggestion.
>
> **Comment 2:** A schematic figure is missing to clearly visualize different components and their relations in the proposed framework (saw some in the appendix but it may be better to put one in the main paper).
>
> **Response:** We will add a schematic figure of our framework, which is condensed from Figure 5, in the main text of our revised manuscript to facilitate the understanding of our framework. Thank you very much.
>
> **Comment 3:** Put some case studies to qualitatively understand the behaviors of the models are missing.
>
> **Response:** Following your suggestions, we will present a case study to illustrate the behaviors of our model in our experiments.
>
> Thank you again for your time and precious advice!

---

> > ### Comment · Reviewer_XKsq · 2022-11-12
> > **Thanks for the rebuttal**
> >
> > I have read the rebuttal. It addressed my comments 2 and 3 but not really 1. What I was asking is some specific real-world examples where the homophily and heterophily around nodes can change along time.
> >
> > I am not modifying my overall evaluating because of the above reasons, but I do think this is a generally solid paper and I am not against its acceptance.

---

> > > ### Author Response · Authors · 2022-11-13
> > > **Clarification for dynamic homophily (dynamic node-wise relationship)**
> > >
> > > Dear Reviewer XKsq,
> > >
> > > Thanks for your follow-up feedback and rephrasing the first question. Actually, the time-varying homophily and heterophily are equivalent to dynamic relations (correlations) between pairwise nodes, thus there are a wide range of scenarios that the homophily and heterophily will change over time, from traffics in the urban road networks to regional air quality in the city. In the road networks, flows at different regions accounting for distinctive functionalities will reveal different patterns at different time intervals, e.g., the tidal patterns force the upstream-downstream flows from residential areas to CBD in the morning while it will be inverse in the evening when people will go back home from their working places. Also, the two regions can reveal the upstream-downstream patterns in the morning while reveal the contended patterns in the afternoon as the context is different, accounting for homophily and heterophily respectively. And for air quality, it is highly related to the wind directions and intensity at different time thus contributing to time-varying region-wise correlations, e.g., the region-wise homophily will vary accordingly if the wind directions have changed. To conclude, the dynamic homophily status in our paper is well-motivated. We will add some reasons of time-varying node-wise correlations in dynamic graphs and modify the Introduction in the next version to enhance the understanding of this phenomenon. Thanks for emphasizing this issue.
> > >
> > > Authors of Paper 1232

---

> > > > ### Comment · Reviewer_XKsq · 2022-11-13
> > > > **Thanks for the further clarifications**
> > > >
> > > > Dear authors,
> > > >
> > > > Thanks for the detailed examples. They make good sense to me. I have raised my overall score to 7.
> > > >
> > > > -- edited: actually there is no option of 7, so I raised it to 8 :)

---

> > > > > ### Author Response · Authors · 2022-11-13
> > > > > **Thanks for appreciation**
> > > > >
> > > > > It is our great pleasure to receive your appreciation, many thanks. We will continue to improve our manuscript until the final version, for better qualifying this highly selected community.

---

### Official Review · Reviewer_mrA2 · 2022-10-31

**Confidence:** 3
**Correctness:** 4
**Technical Novelty And Significance:** 3
**Empirical Novelty And Significance:** 3
**Recommendation:** 8

**Clarity, Quality, Novelty And Reproducibility:**

I have no major concerns with respect to quality, reproducibility, and clarity, and I appreciate its novelty.

**Strength And Weaknesses:**

Strength: 1. The motivation of this paper is good. Topology-task discordance issue does exist in dynamic graph yet has been overlooked. 2. The derived theory and proposed method are sound. 3. Experiments are coherent with the motivation, which clearly demonstrates the effectiveness of the proposed method.

Weakness: This paper is a little bit hard to follow due to a lot of concepts and the complexity of the method. I think the authors could motivate each of the proposed module a little before detailed description. To be specific, it would be better to add summary at the beginning of subsection 5.1 and 5.2.

Minor comments:

The authors should carefully check their citing format- \citet{} and \citep{} are different and should be used accordingly.


**Summary Of The Paper:**

This work studies regression task on dynamic graph. It points out that some edges could be harmful when using GNN for the target task, which the authors called topology-task discordance. To address this, the authors extend the notion of homophily to dynamic graph and design GReTo, which adaptively perform selective message passing on dynamic graph. The extensive experiments showed the effectiveness of the proposed GReTo, which outperforms baselines by a large margin.

**Summary Of The Review:**

I think this is a technically solid, modest-to-high impact paper in a subarea of graph machine learning, so I tend to accept.

---

> ### Author Response · Authors · 2022-11-06
> **Response to Reviewer mrA2**
>
> Dear Reviewer mrA2,
>
> &emsp; Many thanks for your appreciation on our work! We will carefully revise our paper by accepting all your valuable suggestions.
>
> **Comment 1:**  Add summary at the beginning of subsection 5.1 and 5.2.
>
> **Response:**
> 1. Summary of Section 5.1: Given the intuition that different types of neighbors play distinctive roles during the aggregation process, in this subsection, we provide a theoretical analysis on aggregation capacity improvement by extending homophily neighbors to both homophily and heterophily compositions with different kernels. Then we design a signed target-oriented message passing which consists of three stages, 1) target-homophily neighborhood discovery, 2) inter-graph homophily estimation, and 3) disentangle message passing directions and construct aggregation kernels.
>
> 2. Summary of Section 5.2: Since the propagation step is a discrete value, it is intractable to directly optimize such discrete values. In this subsection, we exploit the local neighborhood environments to quantify the informativeness of node representations at each propagation step, which determines when to stop the propagation in a soft manner. Concretely, our high-order propagation is with two components, an adaptive layer importance measure and high-order propagation blocks. We will first introduce the following observations to motivate our designs.
>
>
>
> **Comment 2:**  The citing format may not be correct.
>
> **Response:** We have replaced all ‘\cite{}’ with the ‘\citep’ (as you recommended) in our revised manuscript.
>
> &emsp; Thanks again for your insightful suggestions and we will try our best to further polish our paper to make it better qualify this highly selected community.

---

### Official Review · Reviewer_bmba · 2022-11-01

**Confidence:** 3
**Correctness:** 3
**Technical Novelty And Significance:** 2
**Empirical Novelty And Significance:** 2
**Recommendation:** 6

**Clarity, Quality, Novelty And Reproducibility:**

The notation and writing is hard to follow, and intuition is not clear.

Eq 1: it is not clear for me why this option, and the cases might overlap as the format is different, the middle case looks at the norm of difference whereas the other two at difference of norms. Can you explain what the direction of proximity is?

Also what the inter homophily is defined in eq 1 but not discussed much, why the signs do not match with the intra?

what is d_i in LNE_v_i?

Why figure 5 is in the appendix?


**Strength And Weaknesses:**

The graphs studied are small and the model requires and n^2 step which makes it not scalable to large graphs.

**Summary Of The Paper:**

This paper presents a novel GNN model for dynamic graphs which considers spatial (intra graph: in the same timeframe) and transition (inter-graph: between different timeframes) homophiles by adaptive neighbourhood aggregations and high-order layer propagations. Averaging both gives the target oriented neighbours. The proposed method is called GReTo which empirically gives superior performance in graph regression.

**Summary Of The Review:**

This paper provides a GNN for dynamic graphs which performs well on graph regression. As is, it is hard to follow and understand.

---

> ### Author Response · Authors · 2022-11-06
> **Response to Reviewer bmba (1/2)**
>
> Dear Reviewer bmba,
>
> &emsp; Thanks for your valuable comments to further improve our manuscript! Here we will elaborately address your concerns as follows.
>
> **Comment 1: The graphs studied are small and the model requires n^2 step which makes it not scalable to large graphs.**
>
> **Response:** Thanks for highlighting the efficiency issue. This work focuses on the regression tasks of dynamic graphs concerning urban road network or city networks. At this time, the computation of node-wise spatial proximity is acceptable at the magnitude of hundreds of nodes. When the graphs become larger, the spatial proximity will only be computed on spatially neighboring nodes with a limited computation cost. Assuming there are averagely $N_s$ neighbors of each node, the total computation on a large graph is $N \times N_s$. By adding the temporal evolution computation, it only requires totally $N \times (N_s+1)$ times for the whole graph computation that is linearly increasing with node numbers. Further, we can also construct the matrix describing the ego-net neighbors to allow the computation can run parallelly. We will add some discussions on the efficiency of large graphs in our submission, thanks.
>
> **Questions:**
>
> **Q1: The notations and writing are hard to follow, and the intuition is not clear.**
>
> **Response:** We will check through the notations and polish our words/writing thoroughly.
> The intuition of this work is that not all neighbors are beneficial to aggregate for achieving targets, especially in dynamic graph regression tasks, but only target-oriented (beneficial) neighbors are encouraged to be aggregated. The reason can be lied in that some topologically connected neighbors are dissimilar with targets and the involvement and aggregation of these target-deviated neighbors will introduce noise and harm the performance. Thus, we highlight the beneficial nodes according to fine-grained analysis of neighborhood compositions and further design layer-wise importance measures to propagate over high-order neighborhoods. We will polish our notations, motivations in the revised version.
>
> **Q2: In Eq 1, the cases might overlap as the format is different, the middle case looks at the norm of difference whereas the other two at difference of norms. Explanation on the direction of proximity?**
>
> **Response:** We will clarify this concern on two aspects.
>
> 1. **Clarification on overlap cases of homophily.** The middle case captures the most proximal neighbors without considering the direction, and the other two cases are considered as the deviated neighbors. In these cases, most proximal neighbors do not require the direction indicators as they are considered as the homogenous neighbors for aggregations, while the deviated neighbors are required to impose the signs (positively deviated or negatively deviated) for different kernel-based aggregations as the different signed correlations reveal different patterns in aggregations. Since the ranges of signed distance are divided by $+\varepsilon$ and $-\varepsilon$, there is no overlap or redundancy. More intuitively, the illustration of this design is figured out in the top subfigure of Figure 6(a).
>
> 2. **Direction of the proximity.** The direction of the proximity can be viewed as the sign of difference between the selected node and compared neighbors. Noted that we have exchanged the subscript of intra-homophily calculation and elaborate in **Q3**. In spatial view, the direction indicates how will the node value change when the original node aggregates this compared neighbor. In temporal view, the direction demonstrates how the node is expected to change (increase or decrease) to reach the target. The spatial and temporal directions respectively account for the current graph states and expected evolving states during learning process thus they can be exploited to facilitate predictions. More illustratively, for instance, given intra-graph homophily $(\pi _{ij}^s)_t > 0$, it manifests that if node $v_j$ is aggregated to $v_i$ $(v_i \leftarrow v_j)$, the node value of $v_i$ will increase in the next step and vice versa.  And given inter-graph temporal homophily $(\pi _{ij}^T)_t > 0$, it refers to that $v_i$ is expected to increase its value at the step $t$ by receiving messages from its neighbors to finally obtain the target of $x_j^{t+1}$.
>
> Note that we assume all features $\bf{X}$ are positive, i.e., $\bf{X}>0$ and the norm is to compute the absolute values when the features are multi-dimensional.

---

> > ### Comment · Reviewer_bmba · 2022-11-08
> > **Confirming the responses**
> >
> > Thanks you for the detailed response. I think some of the explanations in these responses can be added to the paper to make it easier to follow and the paper needs heavy polishing, but this can be done in the camera ready version so I am happy to change my score to accept.

---

> > > ### Author Response · Authors · 2022-11-09
> > > **Thanks for follow-up feedback**
> > >
> > > Dear Reviewer bmba, we are appreciated and encouraged to receive your follow-up feedback. Many Thanks. We will continue to working on improving our manuscript to make it as clear as possible. If you have any further questions or concerns, please feel free to let us know.
> > >
> > >
> > >
> > > Authors of Paper 1232

---

> ### Author Response · Authors · 2022-11-06
> **Response to Reviewer bmba (2/2)**
>
> **Q3:The inter homophily is defined in Eq.1 and why the signs do not match with the intra?**
>
> **Response:**
>
> 1. **Definition of inter-homophily.** The inter-graph homophily is defined as the node-wise proximity between adjacent temporal steps ($x_i^t \to x_j^{t + 1}$) in the dynamic graphs (in Line 10 of Section 3.1), which is also illustrated in the bottom-right subgraph of Figure 6 (a).
>
> 2. **The sign mismatching issue.** Thank you for pointing this out. This is a typo on the left part of Eq.(1) and we have exchanged the positions of $i$ and $j$ in the intra-graph homophily and let them consistent with each other, i.e.,
>
> $(\pi_{ij}^s)_t={\begin{array}{*{20}{c}}{ - 1,\frac{{||x_j^t|| - ||x_i^t||}}{{||x_i^t||}} <- \varepsilon }\\\\{\varepsilon, \frac{{||x_j^t - x_i^t||}}{{||x_i^t||}} \le \varepsilon }\\\\{1,\frac{{||x_j^t|| - ||x_i^t||}}{{||x_i^t||}} > \varepsilon }\\\\\end{array}}$
>
> **Q4: d_i in LNE_v_i?**
>
> **Response:** $d_i$ is the degree, a topology feature, for node $v_i$ in graph theory.
>
> **Q5: Issue of Figure 5**.
>
> **Response:** Due to the space limitation, we present the framework overview in Appendix. We will add a subfigure, which is condensed from Figure 5, to overview our framework, and further re-organize some sections of this paper to facilitate the reading. Thanks.
>
> Thank you again for your constructive comments to further polish our paper!

---

### Official Review · Reviewer_dBq6 · 2022-11-02

**Confidence:** 3
**Correctness:** 2
**Technical Novelty And Significance:** 3
**Empirical Novelty And Significance:** 2
**Recommendation:** 6

**Clarity, Quality, Novelty And Reproducibility:**

Several questions:
1. About the topology-task discordance.
In recent years, some models (e.g., AGCRN[3]) learned the topology structure from the data even though there is a topology structure, and achieved better performances than those models based on the pre-defined topology. If we learn the topology from the data directly, will the phenomenon of topology-task discordance still exist?

2. About Fig. 1
Why is Fig. 1(c) negatively correlated, isn’t B connected to B’?
AGCRN[3] learns the diversified patterns (including similar, dissimilar, and even contradictory) by a Node Adaptive Parameter Learning module. What is the difference between AGCRN and GReTo?

3. About Definition 1
This paper categorizes the homophily into three classes, however, as the authors said in the Introduction, for regression tasks, only considering discrete labels will not perform well (see the second paragraph in the Introduction). Then, the homophily is categorized into -1 and 1, is there any information loss?

4. About Definition 2
Why do the authors only consider positive heterophily (1) and negatively heterophily (-1)? Is there no other state, such as 0.3, or 0.5?

5. About Eq. (8)
Why is the positive part interpreted as a low-pass filter while the negative part as a high-pass filter?

6. About Fig. 5
What are the roles of the two temporal convolution networks, respectively?

**Strength And Weaknesses:**

Strength:
1. This paper is well-written and easy to follow, and the motivation is clear.
2. This paper proposes a dynamic graph homophily theory from the spatial-temporal perspective.

Weakness:
1. In the community, there are many GNN-based models for dynamic graphs, but in the Related Work part of this paper, the authors lack some representative methods (e.g., TAMP-S2GCNets[1], D2STGNN[2]). They should clearly discuss the differences between those models.
2. There lack of efficiency analysis in the experimental part.
3. The proposed model GReTo can only make one-step predictions while multi-step predictions are widely used in the community.

[1] TAMP-S2GCNets:
Yuzhou Chen, Ignacio Segovia-Dominguez, Baris Coskunuzer, Yulia R. Gel: TAMP-S2GCNets: Coupling Time-Aware Multipersistence Knowledge Representation with Spatio-Supra Graph Convolutional Networks for Time-Series Forecasting. ICLR 2022
[2] D2STGNN：
Zezhi Shao, Zhao Zhang, Wei Wei, Fei Wang, Yongjun Xu, Xin Cao, Christian S. Jensen: Decoupled Dynamic Spatial-Temporal Graph Neural Network for Traffic Forecasting. Proc. VLDB Endow. 15(11): 2733-2746 (2022)
[3] AGCRN:
Lei Bai, Lina Yao, Can Li, Xianzhi Wang, Can Wang: Adaptive Graph Convolutional Recurrent Network for Traffic Forecasting. NeurIPS 2020

**Summary Of The Paper:**

Aiming at the problems of the aggregations of target deviated neighbors on dynamic graphs (the authors call it topology-task discordance), this paper revisits node-wise relationships based on a dynamic homophily theory from the spatial-temporal perspectives, and proposes a model GReTo consisting of two stages, signed target-oriented message passing and personalized high-order layer propagation, corresponding to the two major processes of GNN.

**Summary Of The Review:**

Please refer to the weaknesses and questions for the details. I look forward to your answers to the above questions.

---

> ### Author Response · Authors · 2022-11-06
> **Response to Reviewer dBq6 (1/3)**
>
> Dear Reviewer dBq6,
>
> Many thanks for your helpful and constructive comments! We will carefully address your concerns and questions in detail.
>
> **W1.Literature review.**
> In this work, we propose a novel perspective to understand the property of dynamic graph-structured data, and devise GReTo to remedy the topology-task discordance for improving spatiotemporal learning performances. Actually, the motivations and graph learning strategy of our work are totally different from your refereed three models. In particular, D2STGNN is motivated by separating the diffusion proportion and inherent traffic information and then designs an estimation gate and a residual decomposition mechanism to model them. For TAMP-S2GCNET, it is inspired by the regularity of time-conditioned topology data and then exploits the dynamics of topology data to capture the shape properties of the complex graph data. In contrast, our work is distinguished from others by 1) GreTo is motivated by the homophily theory in capturing ‘good’ neighbors and addresses the noise involvement induced by topology-task discordance during message passing.  2) Technically, GReTo takes the potential future temporal evolution into account and learns the evolution direction to guide selecting most beneficial nodes. We also perform personalized propagation over high-order neighborhoods with an importance layer measure which has not been studied before. We will carefully discuss these distinctions in the Related Work, and cite these pieces of literature in the appropriate positions, thanks.
>
>
> **W2. Efficiency.** We have discussed the efficiency in A.2 (Complexity analysis). Most of the additional computations are lied in the pre-processing stage.
> In terms of training efficiency, we implement the training process and record the training time on each dataset during this rebuttal stage. The detailed implementation time (seconds) for each epoch is as follows.
> | Datasets| Metr-LA| PeMS-Bay |KnowAir | Temperature|
> |---|---|---|---|---|
> |GReTo  | 58 | 122 |	16	| 16 |
> |GCN |  35 |  87 |	11  |	11 |
>
> We have found that our solution will not sacrifice too much computation and training time to trade for performance.
>
> **W3. Multi-step prediction.** Actually, we have performed multi-step predictions in last paragraph of Section 6 (in Figure 3(b)). We illustrate the results on two extensively-studied datasets, Metr-LA and PeMS-Bay, while other two datasets do not have result reports for comparisons on multi-step settings. The results show that our GReTo also has achieved competitive performances on these datasets and outperformed the STGCN (with a similar spatiotemporal learning setting) with a large margin. This manifests our solution is promising to transfer to multi-step settings, and more specific designs for multi-step predictions, e.g., capturing the sequential trend with regularizations, exploiting some context factors (environments out of the graphs like weather and timestamps) to learn the regularity of sequential trend under specific conditions. These ideas will be left as our future work. Many thanks for your kind remind, which inspires us to further explore the technical solution.
>
>
> **Q1.The topology-task discordance.** We argue that the phenomenon of topology-task discordance can exist in any graph dataset, which is the joint property of graph-structured data and prediction tasks, regardless of technical solutions. For AGCRN [3], it proposes an adaptive topology learning structure with learnable node embedding. AGCRN can capture the spatial proximity in a data-driven manner, but it cannot actively identify the temporal evolution for personalized message passing in both immediate and high-order neighborhoods. Therefore, such discordance also exists in AGCRN. Since the temporal evolution is important to neighbor aggregation in predictions of regression tasks, in contrast, GReTo adaptively and actively adjusts the topology according to temporal evolution directions to aggregate target-beneficial neighbors, and then perform personalized high-order message passing to remedy the topology-task discordance. We will carefully discuss these distinctions in the Related work. Thanks.

---

> > ### Comment · Reviewer_dBq6 · 2022-11-10
> > **Thanks for your response**
> >
> > Thanks for the effort to address my comments. Your clarifications did improve my understanding. It would be better to investigate the training complexity and parameter volumes of the proposed model empirically. Overall, I suggest the authors improve the paper presentation in the final version as, personally, I find the paper hard to follow. I will raise my score to accept.

---

> > > ### Author Response · Authors · 2022-11-10
> > > **Thanks for follow-up feedback**
> > >
> > > Dear Reviewer dBq6,
> > >
> > > Thank you for your follow-up feedback and we are really appreciated for your recognition of our work. The parameter volumes of the proposed model are  both 0.38M on Metr-LA and PeMS-Bay, while the that of MTGNN is approximately 0.41 M and 0.57 M on respective two datasets. According to empirical studies, we can find that our GReTo can be comparably efficient with other solutions where we attach a Table as below to illustrate the comparisons among several models. Noted that the volumes of learnable parameters are partially determined by some hyperparameters like propagation layers, kernel number in each layer.
> > >
> > > Table. Parameter volumes on two datasets
> > > | Model | Metr-LA| PeMS-Bay |
> > > | --------- | ------ | ------ |
> > > | GWN      | 0.28M | 0.28M |
> > > | DCRNN  | 0.37M | 0.37M |
> > > | GRETO  | 0.38M  | 0.38M |
> > > | MTGNN |  0.41M  | 0.57M |
> > >
> > > We will still work on our manuscript to improve the overall quality. Also, more motivation descriptions, distinctions from related works, and complexity analysis will be added into our revised version. Thanks again for your precious suggestions.
> > >
> > > Authors of Paper 1232.

---

> ### Author Response · Authors · 2022-11-06
> **Response to Reviewer dBq6 (2/3)**
>
> **Q2. Explanations on Fig1(c) and difference from AGCRN**
>
> **1) Explanation on Fig1(c) and negatively correlation.**  In the traffic forecasting scenario of Fig1(c), B and B’ are located parallelly between a popular Origin-destination points of O and D, indicating that there must be large-scale human mobilities transfer from O to D. In this case, O-B-D and O-B’-D are two routes contended for traffic flows with competitive relationship. Given the flow volumes transiting from O to D, the flow increasing of B will lead to decreasing of B’. Therefore, B and B’ are negatively correlated.
>
> **2) Difference distinguished from AGCRN.** AGCRN exploits the learnable embedding to capture the diversified patterns between nodes but still cannot identify the positive and negative correlations with future targets awareness, i.e., AGCRN employs the ReLU function to generate all non-negative values in the current graphs, but GReTo can capture the relationships in both positive and negative signs guided by future temporal evolution directions. Therefore, the two distinctions lie in the signed aggregation and temporal evolution awareness.
>
> **Q3.Any information loss on -1 and 1.** The three classes of homophily are proposed to perform a fine-grained analysis on neighborhood distribution for each node and guide the aggregation process, and it has been exploited in the process of aggregation kernel generation $\widehat M_{ij}=\widehat{\pi {_{ij}}^T}\varnothing\pi _{ij}^s$.
>
> During the message passing (node aggregations), each node is aggregated with different weights measured by $\widehat M_{ij}$. Therefore, there is no information loss. For your question, actually, the homophily is exploited as an intermediate result to guide the regression task where the final regression is also employed with the MAPE loss in the second term of Eq (13).
>
> **Q4.Consideration of other states except 1 and -1.** The explanation of this operation and significance is stated before Eq (5) in Target-homophily neighborhood discovery of Section 5.1. We design a three-class homophily quantification to facilitate the fine-grained neighborhood analysis {+1, $\varepsilon$, -1}. The relative values have been considered in the division-based equation $\widehat M_{ij}=\widehat{\pi {_{ij}}^T}\varnothing\pi _{ij}^s$, carrying informative edge information, where each edge value not only discriminates the target-oriented nodes (with positive signs) as good neighbors, but also measures the degree of node-wise proximity where larger values imply more target-oriented consistency.
>
> **Q5.Explanations on low-pass filter and high-pass filter over graph signals.** This is a spectral perspective understanding on graph signals and GNN-based aggregation. The operation of GNNs, i.e., updating information from neighbors, can be seen as a special form of low-pass filter (Wu et al. 2019; Li et al. 2019). We can borrow the theory in spectral domain to explain the graph signals. In particular, the neighboring nodes with similar features (labels) can be viewed as low-frequency signals over graphs while neighboring nodes with dissimilar features (labels) can be viewed as high-frequency proportions over graphs. Actually, the low-frequency composition accounts for  the signals changes smoothly and high-frequency composition accounts for signals changes sharply. In this way, GNNs aggregating the similar neighboring nodes, positively correlated nodes with positive adjacency in $\bf{A}$, can be considered as the low-passing filtering while GNNs aggregating dissimilar neighbors, negatively correlated nodes with negative adjacency in $\bf{A}$, are high-passing filters. Also, this theory has been proved in [5] and widely accepted/exploited in [6,7].

---

> ### Author Response · Authors · 2022-11-06
> **Response to Reviewer dBq6 (3/3)**
>
> **Q6. Roles of the two temporal convolution networks.** This is a sandwich structure of spatiotemporal data mining which is widely-utilized in spatiotemporal learning, e.g., STGCN [4]. This module can capture the temporal evolution where our solution is mostly focusing on graph structure exploitation. This can also help better compare with other solutions with the same setting such as STGCN.
>
> We will carefully incorporate these new results and refined statements in the revised submission. Thank you again for your valuable time and suggestions!
>
> Additional References:
>
> [4] Spatio-temporal graph convolutional networks: a deep learning framework for traffic forecasting. IJCAI 2018.
>
> [5] Beyond Homophily in Graph Neural Networks: Current Limitations and Effective Designs, NIPS 2020.
>
> [6] Beyond Low-frequency Information in Graph Convolutional Networks, AAAI 2021.
>
> [7] Beyond Homophily: Structure-aware Path Aggregation Graph Neural Network, IJCAI 2022.

---

### Author Response · Authors · 2022-11-11
**Summary of revision**

Dear Reviewers and Area Chair,

We thank all of your efforts and valuable time on our manuscript. We are encouraged that reviewers find our work is well-motivated (dBq6, mrA2), the paper is well-written (dBq6,1rTB) and the theoretical analysis/technique is solid (XKsq, mrA2,1rTB). We address all concerns raised by reviewers and have updated the new version of the draft. Next, we briefly describe how we revise our paper in corresponding to your concerns and questions by sections, where the revised contents are highlighted in blue for ease of reading.

Firstly, we have corrected the typo of the left part of Eq (1) by exchanging the two subscripts of $i$ and $j$ in its numerator, and replace the citation format ‘\cite{}’ with ‘\citep{}’. Some additional experiments and discussions have been added.

**Introduction**

We have enhanced the motivation of addressing the topology-task discordance on dynamic graphs by adding the reason of ‘the existence of time-varying values and different edge types’. (Respond to Reviewer bmba, XKsq) In detail, we also add the description of dynamic correlations influenced by tidal patterns  to enhance the motivation of studying on dynamic homophily , i.e., node-wise relationship is dynamic thus homophily is also time-varying over time. (Respond to Reviewer XKsq)

**Related work**

We have additionally discussed two referred models, D2STGNN and TAMP-S2GCNET and explicitly pointed out the distinction of our GReTo is investigating ‘the homophily theory in dynamic graphs to capture ‘good’ neighbors by explicitly modeling the target-related temporal evolution influences’. The refereed works (AGCRN, D2STGNN, TAMP-S2GCNET) have been appropriately cited and the distinctions have now been clearly clarified. (Respond to Reviewer dBq6)

**Preliminaries**

We have clarified some typos and ambiguous statements.

In Def.1, we have corrected the typos of the intra-graph homophily calculation by exchanging the subscript of $i$ and $j$ in the left part of Eq.(1). Also, we have added the detailed explanation of the signed directions in both ${(\pi _{ij}^s)_t}$ and ${(\pi _{ij}^T)_t}$.

In Def.2, we have added ‘the node degree of node $v_i$’ as the definition of $d_i$. Many thanks to Reviewer bmba!


**Remedy topology-task discordance on dynamic graphs**

We have added a schematic figure of our overall solution in Figure 2 (Respond to Reviewer XKsq), and delete the ‘Transition homophily predictor’ of Figure 2 in the original manuscript and leave the Figure 7 as it is (Respond to Reviewer 1rTB).

Following the suggestion of Reviewer mrA2, we have added two summaries of respective Sec. 5.1 and Sec. 5.2 to characterize the overall organization of each subsection.

We have improved the statements concerning ‘low-pass filter and high-pass filter’ and added a reference in Sec. 5.1 for better understanding (Respond to Reviewer dBq6).


**Experiment**

We have added some more discussions on the ablation study results regarding that different properties of datasets may lead to various module sensitivities and thus performances. More details is discussed in the Appendix A.3.3 (Respond to Reviewer 1rTB).

**Appendix**

We also incorporate substantial discussions and experiments in our Appendix.

First, to facilitate the understanding, we have supplemented the detailed motivation of our problem of topology-task discordance, explaining why we learn the homophily on dynamic graphs and what problem our GReTo exactly resolves. (Respond to dBq6, bmba)

Second, we have conducted a series of additional experiments on evaluating the model stability (providing the standard deviations), robustness on temporal interval resolution (learning on longer length of time intervals), and robustness on graphs with higher temporal node degrees (illustrating results on high inter-graph homophily datasets). We add these results by introducing a new subsection A.3.3. (Respond to Reviewer 1rTB).

Third, we have added some experiments to investigate the training complexity and parameter volumes of the proposed model empirically. The additional results are shown in Table 7 and Table 8. (Respond to Reviewer dBq6). To dispel the concern of Reviewer bmba on the generality for large-scale graph, we have provided a complexity analysis of GReTo on large-scale graphs in the second paragraph of A.3.6 (Complexity analysis).

Fourth, we have also incorporated a new discussion subsection of A.3.7, to elaborately discuss the influences of dataset property on module performances and the model scalability on different scenarios. We believe above additional detailed results and analysis can provide better evaluation and understanding of our work, which further verifies the effectiveness and generality of our solution.

In addition, we have revised the colors of Figure 5 and Figure 7.
Thanks for your valuable advice and suggestions! If you have any questions or concerns, please feel free to let us know.

Best regards.

Authors of Paper 1232

---

### Decision · Program_Chairs · 2023-01-20

**Decision:**

Accept: poster

**Justification For Why Not Higher Score:**

The paper is interesting and the presented idea are nice but the algorithm is not too scalable and so it may not be of interest for a wider audience

**Justification For Why Not Lower Score:**

The idea introduced in the paper is novel and the problem is interesting

**Metareview: Summary, Strengths And Weaknesses:**

The authors propose a new GNN technique for dynamic graphs. The main insight of the paper is to consider both intra graph information(focusing on a single timeframe) and inter graph information (changes between timeframes) to improve the performance of the model. In particular, their approach leverage the fact that changes in the graph are signals of positive or negative homophily between nodes.

The paper is well-written and the idea presented in the paper is neat and novel. Experimentally the authors show that their technique obtains good result.

Overall, the paper is a nice contribution and we suggest the acceptance to ICLR

**Note From Pc:**

if the above contains the word "oral" or "spotlight" please see: "oral" presentation means -> notable-top-5% and "spotlight" means -> notable-top-25%. As stated in our emails, we are disassociating presentation type from AC recommendations